# TARG: Training-Free Adaptive Retrieval Gating for Efficient RAG

## Abstract

Retrieval-Augmented Generation (RAG) improves factuality but retrieving for every query often hurts quality while inflating tokens and latency. We propose Training-free Adaptive Retrieval Gating (**TARG**), a single-shot policy that decides when to retrieve using only a short, no-context draft from the base model. From the draft's prefix logits, TARG computes lightweight uncertainty scores—mean token entropy, a margin signal derived from the top-1/top-2 logit gap via a monotone link, or small-$N$ variance across a handful of stochastic prefixes—and triggers retrieval only when the score exceeds a threshold. The gate is model-agnostic, adds only tens to hundreds of draft tokens, and requires no additional training or auxiliary heads. On NQ-Open, TriviaQA, and PopQA, TARG consistently pushes the accuracy–efficiency frontier: compared with Always-RAG[1], TARG matches or improves EM/F1 while reducing retrieval by 70–90% and cutting end-to-end latency, and it remains close to Never-RAG in overhead. A central empirical finding is that under modern instruction-tuned LLMs the margin signal is a robust default (entropy compresses as backbones sharpen), with small-$N$ variance offering a conservative, budget-first alternative. We provide ablations over gate type and prefix length and use a $\Delta$-latency view to make budget trade-offs explicit.

## 1 Introduction

Large language models (LLMs) demonstrate strong performance on knowledge-intensive generation yet remain susceptible to hallucination, producing fluent but unfounded content when parametric memory lacks the necessary facts (Huang et al., 2025; Farquhar et al., 2024). Retrieval-Augmented Generation (RAG) addresses this problem by consulting an external corpus at inference time, coupling a generator with nonparametric memory to improve factuality and transparency (Lewis et al., 2020; Guu et al., 2020; Wang et al., 2023; Izacard & Grave, 2020; Yu et al., 2024; Lin et al., 2023). However, invoking retrieval for every input increases latency and token usage, and can reduce accuracy when retrieved evidence is noisy or tangential. Longer prompts also impose nontrivial computational costs due to the quadratic scaling of self-attention (Vaswani et al., 2017; Dao et al., 2022), and evidence placed in the middle of long contexts is often under-utilized by current models (Liu et al., 2023). These considerations motivate selective retrieval: deciding *when* retrieval is unnecessary is as important as deciding *what* to retrieve.

Recent work explores conditional retrieval, including forward-looking active retrieval (FLARE) that anticipates upcoming content and re-queries when confidence appears low (Jiang et al., 2023), instruction-tuned regulation with reflection tokens (Self-RAG) (Asai et al., 2024), and corrective actions triggered by evidence quality scoring (CRAG) (Yan et al., 2024). Unlike methods that train control heads or run multi-stage tool loops, our method (*i.e.*, TARG) aims to make a single, training-free decision from prefix logits.

We revisit selective retrieval from a simpler angle: a training-free, single-shot decision of *when to retrieve* that any off-the-shelf language model (LM) can make before full decoding. Our approach, named **T**raining-free **A**daptive **R**etrieval **G**ating (**TARG**), uses a short, no-context draft to read the model's own prefix logits and compute a lightweight uncertainty score. We instantiate three signals that require no training or auxiliary heads: (1) mean token entropy, (2) a margin-based score

---

[1]Always-RAG: retrieve for every query; Never-RAG: never retrieve.

derived from the top-1 versus top-2 logit gap via a monotone link, and (3) a small-$N$ variance from a handful of stochastic prefixes. The gate triggers retrieval only when the score exceeds a fixed decision threshold, adding only tens to hundreds of draft tokens per query and integrating cleanly into existing RAG stacks.

A central empirical observation is that the *choice of signal* matters under modern instruction-tuned LLMs. As backbones sharpen, prefix entropies often compress and may lose discriminative power. In contrast, the top-1/top-2 margin retains dynamic range and correlates with cases where external evidence changes or stabilizes the answer; small-$N$ variance behaves similarly but is more conservative. On NQ-Open (Kwiatkowski et al., 2019; Lee et al., 2019), TriviaQA (Joshi et al., 2017), and PopQA (Mallen et al., 2022), TARG with a margin signal traces favorable accuracy–efficiency fronts relative to both Never- and Always-RAG, achieving low retrieval rates and latency overheads close to the Never baseline. We report results with both a compact generator and a stronger Llama-3.1-8B model and adopt a $\Delta$-latency view that makes budget trade-offs explicit.

In summary, this work makes the following contributions:

- **Training-free gate from prefix logits.** We introduce TARG, a lightweight, model-agnostic policy that decides when to retrieve using uncertainty computed from a short prefix; among simple signals, the top-1/top-2 margin emerges as a robust default under modern instruction-tuned LLMs, with small-$N$ variance as a conservative alternative.

- **Budget-aware calibration and cost framing.** We provide a simple recipe to calibrate the decision threshold to a target retrieval budget and report efficiency using $\Delta$ latency, isolating the incremental cost of retrieval and longer prompts from base decoding.

- **Empirical validation across datasets and backbones.** On NQ-Open, TriviaQA, and PopQA, TARG improves the accuracy–efficiency frontier over Always-/Never-RAG at minimal retrieval; trends hold across gate types and persist when upgrading the backbone.

## 2 RELATED WORK

**Retrieval-augmented generation (RAG).** RAG couples parametric generators with non-parametric memory to improve factuality on knowledge-intensive tasks. For example, REALM exposes external knowledge during pretraining via a differentiable retriever (Guu et al., 2020); RAG and FiD popularize end-to-end training of retriever–reader pipelines that condition generation on retrieved passages (Lewis et al., 2020; Izacard & Grave, 2020). Subsequent variants enhance fusion and rationale use (Wang et al., 2023) and adopt rank-then-rerank strategies (Yu et al., 2024). Recent surveys synthesize these design choices and highlight robustness and cost as persistent challenges: long contexts inflate latency, retrieval noise can degrade accuracy, and budgets must be controlled (Wu et al., 2024; Sharma, 2025). These observations motivate policies that decide *when* to retrieve, not only *what* to retrieve.

**Adaptive and active retrieval.** A growing line of RAG research retrieves conditionally. FLARE performs forward-looking active retrieval by anticipating upcoming content and re-querying when predicted tokens look low-confidence (Jiang et al., 2023). Self-RAG trains an LM with reflection/control tokens to decide when to retrieve and how to critique evidence (Asai et al., 2024). CRAG scores the quality of retrieved passages and triggers corrective actions when evidence is weak (Yan et al., 2024). Very recent systems (e.g., SeaKR, SUGAR) leverage uncertainty probes to modulate retrieval frequency (Yao et al., 2024; Zubkova et al., 2025). While effective, these approaches often add supervision, special control tokens, auxiliary probers, or multi-stage loops that increase engineering complexity and latency. In contrast, our *training-free* approach, TARG, makes a single, up-front decision using only prefix logits from the base model; among simple signals, the top-1/top-2 *margin* emerges as a robust default under modern instruction-tuned LLMs, with small-$N$ variance as a conservative alternative.

**Reasoning–acting interleaving with tools.** Frameworks such as Self-Ask and ReAct interleave reasoning with tool use, allowing an LM to search or retrieve on demand during multi-hop solutions (Press et al., 2022; Yao et al., 2023). DSP composes retrieval and generation with programmatic pipelines (Khattab et al., 2022). These methods can yield strong performance but typically rely

on multi-turn tool calls, bespoke prompting, and orchestration overhead. Our scope is orthogonal: we study a *one-shot* gate that decides whether to retrieve at all before full decoding. The two directions are complementary: TARG can suppress unnecessary retrieval in tool-augmented systems, improving latency without altering downstream planners.

**Uncertainty, calibration, and hallucination detection.** Work on LM confidence studies whether models "know when they know," spanning logit-based, internal-state, and consistency-based signals (Geng et al., 2023). SelfCheckGPT, for example, detects hallucinations via sample inconsistency and hence is conceptually related to our small-$N$ variance signal, though we probe only a short prefix rather than full generations (Manakul et al., 2023). TARG operationalizes intrinsic, low-overhead uncertainty from a brief no-context draft (entropy, margin, variance) to decide whether the expected benefit of retrieval outweighs its cost. Empirically, as instruction-tuned backbones sharpen, prefix entropies compress and over-trigger, whereas the top-1/top-2 margin and small-$N$ variance retain discriminative power: yielding selective, budget-aware retrieval that is easy to retrofit and orthogonal to future improvements in retrievers and rerankers.

## 3 METHOD

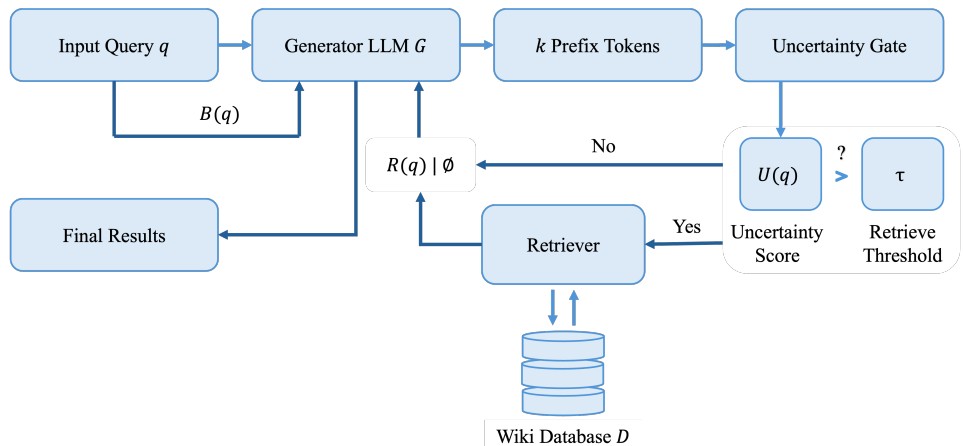

Figure 1: Illustration of TARG methodology

Given a user query $q$ and a generator LLM $G_\theta$ with tokenizer $T$, a RAG system augments the base prompt $B(q)$ with an optional context $C$ retrieved from a corpus $\mathcal{D}$, then decodes an answer $y$. Let $x = B(q) \oplus C$ denote the final prompt (token concatenation), and let the next–token distribution at step $t$ be $p_\theta(y_t \mid y_{<t}, x)$. Our goal is to decide, at inference time and without training, whether to retrieve (i.e., choose $C \neq \varnothing$) for a given $q$ so as to minimize compute cost—retrieval calls, context tokens, and wall-time latency—while preserving or improving task accuracy.

### 3.1 PREFIX UNCERTAINTY FROM A SHORT DRAFT

We exploit the model's own uncertainty on a short, retrieval-free prefix to decide whether retrieval is needed. We run $G_\theta$ for $k$ tokens on the base prompt only, obtaining logits $\ell_1, \ldots, \ell_k$ and probabilities $\pi_t = \text{softmax}(\ell_t), t \in \{1, 2, \ldots k\}$. Each gate yields a per-step score $u_t$, aggregated as $U = \frac{1}{k} \sum_{t=1}^{k} u_t$.

**Entropy gate** Per–step entropy $H_t = -\sum_j \pi_{t,j} \log \pi_{t,j}$, with

$$U_{\text{ent}}(k) = \frac{1}{k} \sum_{t=1}^{k} H_t, \tag{1}$$

so larger $U_{\text{ent}}$ indicates higher uncertainty.

**Margin-as-uncertainty gate**  Let $g_t = \ell_{t,(1)} - \ell_{t,(2)} \geq 0$ be the top-1 vs. top-2 logit gap at step $t$. Map gaps to a positive uncertainty via a strictly decreasing $\phi$ (default $\phi(z) = \exp(-z/\beta)$, temperature $\beta > 0$):

$$U_{\mathrm{mar}}(k;\beta) = \frac{1}{k}\sum_{t=1}^{k}\phi(g_t) \in (0,1]. \tag{2}$$

Because $\phi$ is strictly decreasing, thresholding $U_{\mathrm{mar}}$ is order-equivalent to thresholding the *mean gap*; the link is a convenience rather than a fragile choice.

***Lemma 1 (order-equivalence).*** For any strictly decreasing $\phi$, thresholding $U_{\mathrm{mar}}$ at $\tau$ is equivalent to thresholding the mean gap at some $\tau'$, i.e., decisions are identical up to a monotone reparameterization. (Proof shown in Appendix.)

**Small-$N$ variance gate**  Sample $N$ short stochastic prefixes (temperature $T$) to obtain sequences $s^{(1)}, \ldots, s^{(N)}$. At step $t$, let $\hat{p}_t$ be the empirical token distribution of $\{s_t^{(n)}\}$ and define $d_t = 1 - \max_j \hat{p}_t(j)$. Then

$$U_{\mathrm{var}}(k,N) = \frac{1}{k}\sum_{t=1}^{k}d_t, \qquad 0 \leq U_{\mathrm{var}} \leq \frac{N-1}{N}. \tag{3}$$

***Lemma 2 (boundedness).*** The mode frequency is $\geq 1/N$, hence $d_t \leq (N-1)/N$ and the stated bound. (Proof shown in Appendix.)

### 3.2 GATE DECISION AND DECODING

Given the scalar uncertainty score $U$ from the prefix draft, we trigger retrieval if $U > \tau$, the $\tau$ is the retrieve threshold:

$$\mathrm{retrieve}(q) \iff U(q) > \tau. \tag{4}$$

If retrieval is triggered, we construct the context $C$ with a dense encoder $E$ (e.g., E5-base-v2 (Wang et al., 2022)) and a FAISS inner-product index (Douze et al., 2024). Let $h = \mathrm{norm}(E(q))$ be the normalized query embedding; we return top-$K$ passages $D_K(q)$ by similarity $\langle h, E(d) \rangle$ and format the context as

$$C = \bigoplus_{d \in D_K(q)} \mathrm{format}(d)[1{:}L_{\mathrm{ctx}}], \tag{5}$$

truncated to the context budget $L_{\mathrm{ctx}}$ tokens. The final prompt is $x = B(q) \oplus C$; otherwise (no retrieval) we proceed with zero-RAG using $x = B(q)$. For long generations, an optional single re-check can be applied every $m$ tokens: if the running-prefix score exceeds $\tau$ and retrieval has not yet occurred, retrieve once and continue decoding.

---

**Algorithm 1** TARG inference (training-free gate)

---

**Require:** query $q$, threshold $\tau$, prefix length $k$, (optional) recheck stride $m$, retriever $R$, top-$K$,
    context budget $L_{\mathrm{ctx}}$
  1: $x \leftarrow B(q)$
  2: draft $\leftarrow$ DECODEPREFIX$(G_\theta, x, k)$
  3: $U \leftarrow$ SCORE(draft; gate $\in \{\mathrm{entropy}, \mathrm{margin}, \mathrm{variance}\}$)
  4: **if** $U > \tau$ **then**
  5:     $C \leftarrow$ RETRIEVEFORMAT$(R, q, K, L_{\mathrm{ctx}})$
  6:     $x \leftarrow B(q) \oplus C$
  7: **end if**
  8: $y \leftarrow$ GENERATE$(G_\theta, x)$            ▷ optionally re-check every $m$ tokens if not yet retrieved
  9: **return** $y$

---

In all main experiments in Section 5, we use TARG in its *single-shot* form: we decode a short prefix from the base prompt $B(q)$, compute $U(q)$ once, and then either retrieve exactly once or never retrieve. Algorithm 1 also illustrates an online extension for very long generations in which the gate can be re-evaluated every $m$ tokens and retrieval can be triggered mid-generation if $U(q)$ ever exceeds $\tau$. In our short-answer QA setting, however, a simple re-check policy with a fixed $\tau$

is overly aggressive: as shown in Appendix A.9, it drastically increases retrieval rate and latency without greatly improving EM, effectively degenerating toward Always-RAG. We therefore adopt single-shot gating for all reported results and view dynamic re-check as a design option for future long-form scenarios.

We also explored simple aggregation of the three gates (e.g., normalizing each $U$ to $[0, 1]$ and averaging). On a dev split, these ensembles did not yield consistent improvements over the best single gate once $\tau$ and $k$ are tuned; the combined scores behave like re-parameterized margin gates while introducing extra weights to calibrate. For simplicity and robustness, we therefore present each gate separately and recommend MARGIN as a default.

### 3.3 Cost, accuracy and calibration

**Cost model.** Let $T_{\text{draft}}{=}k$ denote the always-incurred prefix, $T_{\text{ctx}}$ the context tokens when retrieving, and $T_{\text{out}}^{(0)}, T_{\text{out}}^{(1)}$ the output tokens without/with retrieval. With retrieval rate $\pi(\tau) = \Pr[U > \tau]$, the expected LM tokens per query are

$$\mathbb{E}[T(\tau)] \;=\; T_{\text{draft}} + (1 - \pi(\tau))\,\mathbb{E}[T_{\text{out}}^{(0)}] + \pi(\tau)\left(T_{\text{ctx}} + \mathbb{E}[T_{\text{out}}^{(1)}]\right), \tag{6}$$

and an analogous additive decomposition holds for wall-time latency. We report efficiency using $\Delta$ latency to isolate the incremental overhead vs. the zero-RAG baseline.

**Accuracy model and dominance.** Let $A^{(0)}(q)$ and $A^{(1)}(q)$ be correctness indicators (or probabilities) for zero-RAG and with-RAG, and let $\Delta(q) = A^{(1)}(q) - A^{(0)}(q)$. We assume *usefulness calibration* at the level of conditional expectations: there exists a threshold $\tau_*$ such that

$$\mathbb{E}[\Delta(q) \mid U(q) \le \tau_*] \le 0 \quad \text{and} \quad \mathbb{E}[\Delta(q) \mid U(q) > \tau_*] \ge 0. \tag{7}$$

This is an *average-case modeling assumption* about the joint distribution of $(U, \Delta)$ under a fixed retriever/corpus, not a pointwise guarantee and not a statement that must hold for every possible dataset or routing policy. In particular, if retrieval systematically helps when $U$ is small and hurts when $U$ is large (e.g., adversarial or heavily filtered traffic concentrated in such regions), a threshold $\tau_*$ satisfying these inequalities may not exist and our dominance argument would not apply.

Under usefulness calibration, choosing $\tau \approx \tau_*$ yields

$$\mathbb{E}[A_{\text{gate}}(\tau)] = \mathbb{E}[A^{(0)}] + \mathbb{E}[\Delta(q)\,\mathbf{1}\{U(q) > \tau\}] \gtrsim \max\{\mathbb{E}[A^{(0)}], \mathbb{E}[A^{(1)}]\}. \tag{8}$$

*Intuition.* Always-RAG integrates $\Delta$ over all $q$, including negative-$\Delta$ regions where $U \le \tau_*$; Never-RAG integrates zero. Thresholding near $\tau_*$ admits only the positive region. Appendix A.7 ("Error quadrants over $(U(q), \Delta(q))$") provides empirical support that, under our frozen dense retriever over chunked Wikipedia, the learned uncertainty scores are aligned with this average-case assumption.

**Threshold calibration and budget control.** Let $F_U$ be the empirical CDF of $U$ on a development set (denoted by dev). To hit a retrieval budget $\rho \in [0, 1]$, pick $\tau = F_U^{-1}(1-\rho)$. Alternatively, select $\tau$ that maximizes dev accuracy. Because $U$ is scalar, calibration is fast and stable. In practice we tune $k$ (prefix length), $\beta$ for the margin link, and $N$ for variance on the same dev split.

### 3.4 Implementation details

We use a flat inner-product FAISS (Douze et al., 2024) index with a normalized dense encoder; $K$ and $L_{\text{ctx}}$ are tuned on dev. For the margin gate we default to $\phi(z) = \exp(-z/\beta)$ with $\beta{=}1$, which preserves ordering (Lemma 1) and yields interpretable $U \in (0, 1)$. For variance we use $N{=}3$ samples by default (upper bound $2/3$). All gates reuse the same $k$-token prefix, so the gating step adds only a few dozen to a few hundred tokens per query.

## 4 Experimental Setup

**Models, decoding, and baselines.** We evaluate two instruction-tuned backbones, Qwen2.5-7B-Instruct (Bai et al., 2023) and Llama-3.1-8B-Instruct (Grattafiori et al., 2024), under the same decoding protocol: greedy generation (no sampling), batch size 1, identical prompts/stop criteria, and

the same short prefix for gating. The prefix is decoded without retrieval using the same greedy policy; for the variance gate only, we draw $N$ short prefixes at temperature $T=0.7$. We compare Never-RAG (decode from $B(q)$), Always-RAG (retrieve once per query and decode from $B(q) \oplus C$), and our training-free gate TARG (draft $k$ tokens without retrieval, compute a scalar uncertainty $U$, and retrieve if $U > \tau$). All main experiments use a single-shot gate: we decide once based on the initial $k$-token prefix and never perform mid-generation re-checks; a dynamic re-check variant is evaluated in the Appendix A.9.

**Retriever, index, and corpus.** Without losing generality regarding retriever type, which was proven in Appendix A.4, we use a frozen dense dual encoder E5-BASE-V2 (Wang et al., 2022) and follow model guidelines by prefixing queries with `"query:"` and passages with `"passage:"`. Queries and passages are encoded into 768-d vectors, $\ell_2$-normalized, and stored in a FAISS IndexFlatIP (Douze et al., 2024); with normalization, inner product equals cosine similarity. At inference, we return top-$K=5$ passages and concatenate them as the retrieval context (formatted as `[title] text` and truncated to fit a context budget $L_{\text{ctx}}$). The corpus is English Wikipedia (Foundation); articles are chunked into passages of roughly 1000 characters with 100-character overlap (minimum 200 characters) and a large subset is indexed for all experiments. Our aim is to evaluate a training-free gating policy, which emphasizes when to retrieve rather than to optimize the retrieval stack; the gate is orthogonal to retriever quality and can sit atop stronger encoders, rerankers, or context compression modules.

**Datasets and evaluation protocol.** We evaluate on NQ-Open (Lee et al., 2019), TriviaQA (Joshi et al., 2017), and PopQA (long-tail entities) (Mallen et al., 2022). Results are reported for Never-/Always-RAG and gated operating points; for NQ-Open, we include threshold sweeps and ablations. Prefix length is ablated over $k \in \{10, 20, 30\}$, with $k=20$ used by default (best cost/quality balance). We instantiate three training-free signals from the $k$-token prefix: (i) **Entropy** $U_{\text{ent}} = \frac{1}{k} \sum_t \left( - \sum_j \pi_{t,j} \log \pi_{t,j} \right)$; (ii) **Margin** $U_{\text{mar}} = \frac{1}{k} \sum_t \phi(g_t)$ where $g_t$ is the top-1/top-2 logit gap and $\phi(g) = \exp(-g/\beta)$ with $\beta=1$ (order-preserving link yielding $U \in (0, 1]$); and (iii) **Variance** $U_{\text{var}} = \frac{1}{k} \sum_t \left( 1 - \max_j \hat{p}_t(j) \right)$ from $N=3$ short stochastic prefixes at $T=0.7$, with range $[0, (N-1)/N]$ ($N$ is the number of short prefixes).

**Threshold sweeps, metrics, and reproducibility.** For each gate, we sweep the decision threshold $\tau$ to expose accuracy–efficiency frontiers. We report Exact Match (EM) and F1 (standard normalization), Retrieval Rate $\pi = \Pr[U > \tau]$, and $\Delta$ latency — extra seconds per query relative to the Never-RAG baseline on the same dataset/hardware; absolute latencies are provided once in table footnotes. Calibration of $\tau$ can target a retrieval budget via the empirical CDF of $U$ on a `dev` set, but in main results we present grid sweeps and select representative operating points at modest budgets ($\sim$5–20%). Because decoding is deterministic (greedy) and the retriever is frozen, randomness enters only through the variance gate's short prefixes and dataset sampling. To quantify statistical reliability, we compute bootstrap 95% confidence intervals over queries for EM/F1 and, for key configurations (Never-RAG, Always-RAG, and the best TARG operating points), repeat evaluation under different RNG seeds in the Appendix A.8. All runs use batch size 1 and identical decoding parameters across modes; for variance we fix the temperature and RNG seed in main tables and vary them only in ablations.

## 5 RESULTS

Table 1 shows that unconditionally retrieving (ALWAYS) increases latency and often depresses accuracy relative to NEVER, a hallmark of off-topic or aliased passages. Training-free gating restores precision: both MARGIN and VARIANCE match or exceed NEVER while staying far cheaper than ALWAYS. On TriviaQA, MARGIN attains the best quality by retrieving more frequently, whereas VARIANCE achieves essentially the same quality at a tiny retrieval rate and negligible overhead, reflecting its conservative trigger. On PopQA, where retrieval precision is harder, both gates improve over ALWAYS and surpass NEVER with moderate added cost; the gains are modest, consistent with long-tail entity drift. On NQ-Open, small amounts of retrieval help: ENTROPY/VARIANCE achieve slight improvements at very low budgets, while MARGIN trades a bit more budget for similar quality. For intermediate retrieval rates (e.g., $\pi$ between 0.05 and 0.3; see Appendix A.8 for full sweeps),

Table 1: **Selective retrieval with Qwen2.5-7B-Instruct.** Representative TARG operating points (two per dataset) against non-gated baselines. The threshold $\tau$ is shown for gated runs; "–" denotes baselines. $\Delta$ Latency is the added seconds per query relative to the Never-RAG baseline on the same dataset/hardware; the NEVER row shows the absolute baseline latency (s/q) in parentheses.

| Dataset | Model | $\tau$ | EM / F1 (%) ↑ | Retrieval Rate | $\Delta$ Latency (s/q) ↓ |
|---|---|---|---|---|---|
| | NEVER | – | 60.8 / 61.4 | 0.000 | 2.947 (Baseline) |
| | ALWAYS | – | 57.6 / 57.2 | 1.000 | +3.462 |
| TriviaQA | ENTROPY | 0.80 | 61.8 / 62.2 | 0.028 | +0.876 |
| | MARGIN | 0.15 | **62.2 / 62.6** | 0.338 | +2.174 |
| | VARIANCE | 0.75 | 61.8 / 62.2 | 0.006 | **+0.133** |
| | NEVER | – | 20.0 / 20.1 | 0.000 | 2.129 (Baseline) |
| | ALWAYS | – | 14.6 / 14.6 | 1.000 | +3.828 |
| PopQA | ENTROPY | 0.80 | 22.4 / 22.3 | 0.124 | **+1.761** |
| | MARGIN | 0.35 | **23.0 / 23.1** | 0.124 | **+1.761** |
| | VARIANCE | 0.40 | 22.8 / 22.9 | 0.182 | +1.847 |
| | NEVER | – | 38.8 / 37.7 | 0.000 | 3.293 (Baseline) |
| | ALWAYS | – | 37.4 / 36.7 | 1.000 | +2.922 |
| NQ-Open | ENTROPY | 0.85 | **39.6 / 39.1** | 0.046 | +0.964 |
| | MARGIN | 0.20 | **39.6** / 38.8 | 0.304 | +1.295 |
| | VARIANCE | 0.60 | 38.6 / 38.0 | 0.012 | **+0.291** |

Table 2: **Selective retrieval with Llama-3.1-8B-Instruct.** Same protocol as Table 1. $\Delta$ Latency reports added seconds per query over the dataset's Never baseline; absolute Never latencies (s/q) appear in parentheses.

| Dataset | Model | $\tau$ | EM / F1 (%) ↑ | Retrieval Rate | $\Delta$ Latency (s/q) ↓ |
|---|---|---|---|---|---|
| | NEVER | – | 80.8 / 80.0 | 0.000 | 10.383 (Baseline) |
| | ALWAYS | – | 67.6 / 67.2 | 1.000 | +1.069 |
| TriviaQA | ENTROPY | 0.80 | 74.4 / 74.1 | 0.524 | +0.495 |
| | MARGIN | 0.70 | **83.8 / 83.0** | 0.001 | **+0.018** |
| | VARIANCE | 0.75 | 83.6 / **83.0** | 0.001 | **+0.018** |
| | NEVER | – | 35.2 / 34.4 | 0.000 | 10.299 (Baseline) |
| | ALWAYS | – | 24.8 / 24.6 | 1.000 | +1.269 |
| PopQA | ENTROPY | 0.80 | 28.8 / 28.8 | 0.760 | +0.974 |
| | MARGIN | 0.45 | **36.6 / 36.2** | 0.108 | +0.424 |
| | VARIANCE | 0.55 | 36.4 /36.0 | 0.084 | **+0.317** |
| | NEVER | – | 53.8 / 51.7 | 0.000 | 10.299 (Baseline) |
| | ALWAYS | – | 48.6 / 46.1 | 1.000 | +1.248 |
| NQ-Open | ENTROPY | 0.95 | 55.4 / 53.1 | 0.132 | +0.175 |
| | MARGIN | 0.50 | **57.6 / 54.7** | 0.008 | **+0.012** |
| | VARIANCE | 0.60 | 56.8 / 53.7 | 0.026 | +0.059 |

TARG with the MARGIN or VARIANCE gates attains higher EM/F1 than both NEVER and ALWAYS while adding minimal latency, indicating that the gate is not merely imitating the better unconditional baseline but actively identifying when external evidence is likely beneficial. Overall, with a 7B-class model, VARIANCE is a strong default when budgets are tight; MARGIN is preferable when a small accuracy boost justifies higher (but still selective) retrieval.

With a stronger generator (Table 2), the full frontier shifts upward while the gate ordering becomes clearer. ENTROPY now over-triggers and lags in quality, indicating that prefix entropies compress under sharper models and lose discriminative range. In contrast, MARGIN and VARIANCE achieve near-best accuracy at vanishing retrieval budgets and essentially zero overhead (e.g., +0.012 s on

Table 3: **External reference systems (context only).** Results reported by prior work on our evaluation datasets. These systems are *not directly comparable* to our training-free, frozen-retriever setup: most use larger backbones and/or trained retrievers/rerankers and may differ in corpora and scoring. Values are EM or EM/Acc, whichever available.

| Models | NQ EM | TriviaQA EM | PopQA EM |
|---|---|---|---|
| *Without Retrieval-Augmented Generation* | | | |
| GPT-4-0613 (OpenAI, 2024) | 40.3 | 84.8 | 31.3 |
| GPT-4-turbo-2024-0409 (OpenAI, 2024) | 41.5 | 80.0 | 25.0 |
| *With Retrieval-Augmented Generation* | | | |
| FiD-Large (Izacard & Grave, 2020) | 51.4 | 61.6 | – |
| RFiD-Large (Wang et al., 2023) | 54.3 | 72.6 | – |
| RA-DIT 65B (Lin et al., 2023) | 35.2 | 75.4 | – |
| Llama3-RankRAG 8B (Yu et al., 2024) | 50.6 | 82.9 | 57.6 |
| Llama3-RankRAG 70B (Yu et al., 2024) | 54.2 | **86.5** | **59.9** |
| Qwen2.5 7B-Entropy ($\tau = 0.85, 0.8, 0.8$) | 39.6 | 61.8 | 22.4 |
| Qwen2.5 7B-Margin ($\tau = 0.2, 0.15, 0.35$) | 39.6 | 62.2 | 23.0 |
| Qwen2.5 7B-Variance ($\tau = 0.6, 0.75, 0.4$) | 38.6 | 61.8 | 22.8 |
| LLAMA3.1 8B-Entropy ($\tau = 0.95, 0.8, 0.8$) | 55.4 | 74.4 | 28.8 |
| LLAMA3.1 8B-Margin ($\tau = 0.5, 0.7, 0.45$) | **57.6** | 83.8 | 36.6 |
| LLAMA3.1 8B-Variance ($\tau = 0.6, 0.75, 0.55$) | 56.8 | 83.6 | 36.4 |

NQ-Open), because the top-1/top-2 logit gap and small-$N$ disagreement retain spread even when the distribution is globally peaked. Practically, use MARGIN by default; use VARIANCE when budgets are extremely tight. The very small retrieval rates (e.g., $\pi \approx 0.001$) in these settings should be viewed as extreme-budget points that illustrate the shape of the accuracy–efficiency frontier; more moderate budgets (e.g., $\pi$ between $0.05$ and $0.3$, see Appendix A.8) show similarly robust gains and are more relevant for deployment.

Table 3 situates our numbers against widely reported results that vary in backbone size, trained retrieval/reranking, corpora, and scoring. These entries indicate headroom rather than serve as baselines. Two messages follow. First, absolute scores scale strongly with backbone and retrieval engineering. Second, selective retrieval is orthogonal to those choices: a calibrated, training-free gate can be dropped into stronger stacks and should continue to reduce unnecessary retrieval and latency.

Across datasets and backbones, the tables reveal a consistent picture. First, ALWAYS is not a safe default: when the top-$K$ set contains distractors or aliases, small and mid-size generators spend compute on irrelevant context and drift, so accuracy falls while latency rises. The effect is most visible on long-tail entity queries (e.g., PopQA), where retrieval precision is intrinsically harder. Second, the choice of uncertainty signal is pivotal and interacts with backbone sharpness. As instruction-tuned LMs become more peaked, prefix entropies compress and lose ranking power; in contrast, the top-1/top-2 logit gap (MARGIN) and small-$N$ disagreement (VARIANCE) retain dynamic range and better correlate with cases where external evidence flips or stabilizes the answer. In Appendix A.6, we quantify this by binning queries by the gate score $U(q)$ and plotting base-model accuracy, observing a clear monotone decrease as uncertainty increases for the MARGIN and VARIANCE gates. Appendix A.7 further provides a quadrant-based error analysis over $(U(q), \Delta(q))$ that highlights where retrieval is beneficial, neutral, or harmful, offering empirical support for the usefulness-calibration assumption in Section 3.3.

This explains why, under Llama-3.1–8B, MARGIN/VARIANCE achieve near-best quality at vanishing retrieval rates and essentially zero added wall-time, while ENTROPY over-triggers and underperforms. Third, efficiency should be interpreted as a *budget*, not just absolute time. Reporting $\Delta$ latency isolates the incremental cost of retrieval and longer prompts beyond base decoding; the strongest MARGIN/VARIANCE operating points cluster near the NEVER baseline in latency yet exceed ALWAYS in accuracy, yielding a controllable and deployment-friendly accuracy–efficiency frontier. Finally, these gains are backbone-agnostic: with both Qwen2.5–7B and Llama-3.1–8B,

TARG improves the frontier; the preferred gate depends on budget and model sharpness—use MAR-GIN by default, VARIANCE when budgets are extremely tight, and reserve ENTROPY for ablations or weaker backbones.

In deployment, calibrate the gate on a development set to match a target retrieval budget (via the empirical CDF of the gate score). Use MARGIN by default with modern instruction-tuned LLMs; switch to VARIANCE when budgets are extremely tight. Keep ENTROPY as an ablation or for weaker backbones. Finally, report accuracy together with retrieval rate and $\Delta$ latency so the quality–cost frontier is explicit rather than implicit.

# 6 DISCUSSION

**Selective retrieval vs. unconditional retrieval.** Across datasets and backbones, unconditionally retrieving (ALWAYS) is not a safe default. When the top-$K$ set contains distractors or aliases, the generator spends compute integrating irrelevant context; longer prompts further disperse attention, increasing the chance that salient evidence is ignored. These effects are most acute on long-tail, entity-heavy queries (e.g., PopQA), where lexical/semantic similarity can be misleading. In contrast, a training-free gate that abstains on easy, high-confidence cases and triggers on genuinely uncertain ones raises the *precision* of retrieval: prompts stay shorter on average, latency remains close to NEVER, and accuracy does not suffer from "anchoring on a distractor." This explains why ALWAYS pays a consistent latency tax while often underperforming NEVER.

**Which uncertainty signal—and why.** Backbone sharpness governs the usefulness of different prefix signals. As instruction-tuned models become more peaked, prefix entropies compress and lose ranking power—entropy can be low even when the model is confidently wrong—so ENTROPY tends to over-trigger and underperform. By contrast, the top-1/top-2 logit gap retains dynamic range: genuine ambiguity shrinks the gap, raising a margin-based score; small-$N$ variance captures a similar phenomenon via disagreement across a few stochastic drafts. Empirically, MARGIN and VARIANCE achieve near-best accuracy at vanishing retrieval rates and essentially zero added wall-time under stronger backbones (e.g., Llama-3.1-8B), while ENTROPY lags. A practical rule emerges: use MARGIN by default; switch to VARIANCE when budgets are extremely tight; keep ENTROPY mainly for ablations or weaker backbones.

**Dataset effects and PopQA.** Selective retrieval is most effective when (i) many queries are already covered by parametric knowledge and (ii) occasional retrieval injects noise or redundancy—conditions that hold for NQ-Open and TriviaQA. PopQA stresses long-tail entities, aliases, and temporally sensitive facts; dense retrieval can surface near-misses with high similarity that distract the generator. Under our deliberately training-free stack (frozen encoder, modest $K$), absolute headroom is limited, but the gate still improves the quality–cost frontier relative to ALWAYS/NEVER. Orthogonal upgrades—hybrid BM25+dense retrieval, cross-encoder reranking, entity-aware scoring—raise absolute accuracy across all modes; a calibrated gate continues to suppress wasteful retrieval on easy queries and preserves efficiency as the stack improves.

**Budgets, not just time.** We report $\Delta$ latency as added seconds per query over NEVER on the same hardware, isolating the incremental cost of prefix drafting, retrieval, and longer-context decoding from base decoding: $\mathbb{E}[\Delta t] \approx t_{\text{draft}} + \pi(\tau)\, t_{\text{retrieval}} + \pi(\tau)\, t_{\text{decode|ctx}}$. The strongest MARGIN/VARIANCE operating points cluster near the NEVER baseline in $\Delta$ latency yet exceed ALWAYS in accuracy, yielding a controllable and deployment-friendly accuracy–efficiency frontier. In practice, calibrate the threshold to a target retrieval budget by matching the empirical CDF of gate scores on a development set.

**Context against larger systems.** Reference numbers from training-heavy stacks (Table 3) provide headroom, not baselines: they differ in backbone size, trained/hybrid retrieval and reranking, corpora/snapshots, and scoring. Our contribution is orthogonal—deciding *when* to retrieve—and can be dropped into stronger stacks: as retrieval/reranking improves, all curves shift upward while a calibrated gate continues to avoid unnecessary retrieval on easy queries.

# 7 LIMITATIONS

Our study targets a practical and narrowly defined question—how to decide *when* to retrieve—so several aspects are intentionally scoped. However, there are still limitations together with straightforward paths for us to improvement: (i) We evaluate English open-domain QA over Wikipedia. While this setting is standard and stresses retrieval precision, it does not cover domain-specific corpora, or multilingual inputs. Future work can apply the same training-free gate to hybrid or domain corpora (e.g., web, scientific papers) and multilingual encoders, and to tasks such as fact verification or long-form answer drafting. (ii) TARG uses a single threshold tuned on a small development set. Although calibration is fast and stable (via the empirical CDF), thresholds may shift with models, prompts, or domains. (iii) We deliberately use a frozen dense encoder to isolate the "when-to-retrieve" decision. Absolute headroom is bounded by retrieval precision. This is orthogonal to our method: the same gate can ride on stronger stacks (such as cross-encoder reranking). A compact future experiment is to show that as retrieval improves, all modes rise while TARG continues to suppress unnecessary retrieval.

# 8 CONCLUSION

We revisited retrieval-augmented generation from a simple angle: *deciding when to retrieve* with no additional training. The proposed TARG policy reads uncertainty from a short, retrieval-free prefix and triggers retrieval only when warranted. Among training-free signals, the MARGIN score—derived from the top-1/top-2 logit gap—emerges as a robust default under modern instruction-tuned backbones; VARIANCE provides a conservative alternative when budgets are extremely tight. Framed through $\Delta$ latency and retrieval rate, TARG consistently shifts the accuracy–efficiency frontier: it matches or exceeds NEVER while avoiding the accuracy and latency penalties of ALWAYS, and it does so at vanishing retrieval budgets. In practice, deploying TARG requires only a lightweight one-dimensional calibration of the threshold $\tau$ (and optionally the prefix length $k$) on a small development set, in contrast to methods that train auxiliary heads or control tokens; our sensitivity analysis shows a broad region of good $(k, \tau)$ settings rather than a narrow optimum.

Beyond raw numbers, the results deliver two actionable insights. First, unconditional retrieval is not a safe default: when top-$K$ contains distractors or aliases, longer prompts and off-topic context hurt both quality and latency. Second, backbone sharpness dictates which uncertainty signal discriminates: prefix entropies compress under stronger models, whereas the logit gap and small-$N$ disagreement retain dynamic range and remain predictive of when external evidence will flip or stabilize the answer. These observations turn a one-line threshold into a practical control knob: set the budget you can afford, calibrate once, and deploy.

TARG is intentionally plug-and-play. It neither assumes trained probers nor requires changes to the retriever; it adds only tens to hundreds of draft tokens and introduces a single scalar control. As retrieval stacks improve (hybrid search, reranking, better chunking), absolute accuracy rises while the gate continues to suppress wasteful retrieval on easy queries—yielding similar or larger efficiency dividends at higher quality. We view training-free gating as a basic primitive for RAG systems: a small, dependable mechanism that restores precision to retrieval, makes latency predictable, and is simple enough to be widely adopted.

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

## A APPENDIX

### A.1 PROOF OF LEMMA 1 (ORDER-EQUIVALENCE FOR THE MARGIN GATE)

Recall the per-step logit gaps $g_t = \ell_{t,(1)} - \ell_{t,(2)} \geq 0$ for $t = 1, \ldots, k$ and a strictly decreasing, continuous *margin link* $\varphi : \mathbb{R}_{\geq 0} \to (0, 1]$ (e.g., $\varphi(z) = e^{-z/\beta}$). Define

$$U_{\mathrm{mar}}(k; \varphi) \triangleq \frac{1}{k} \sum_{t=1}^{k} \varphi(g_t), \qquad \text{and retrieve if } U_{\mathrm{mar}} > \tau.$$

Fix any "shape" vector $\boldsymbol{\delta} = (\delta_1, \ldots, \delta_k)$ with $\sum_t \delta_t = 0$ and consider the family of gap vectors $\mathbf{g}(\mu) = \mu \mathbf{1} + \boldsymbol{\delta}$, parameterized by the mean gap $\mu = \frac{1}{k} \sum_t g_t$.

**Lemma 1** (Location-equivalence of the margin gate). *Let $\phi : \mathbb{R} \to \mathbb{R}$ be strictly decreasing. Fix any "shape" vector $\delta = (\delta_1, \ldots, \delta_k) \in \mathbb{R}^k$ with $\sum_{t=1}^{k} \delta_t = 0$, and define the gap sequence along the location family*

$$g(\mu) = (\mu + \delta_1, \ldots, \mu + \delta_k) \in \mathbb{R}^k, \qquad \mu \in \mathbb{R}.$$

*Define the margin-based gate statistic*

$$U_{\mathrm{mar}}(\mu) = \frac{1}{k} \sum_{t=1}^{k} \phi(\mu + \delta_t).$$

*Then $U_{\mathrm{mar}}(\mu)$ is strictly decreasing in $\mu$. Consequently, for any threshold $\tau \in \mathbb{R}$ there exists a (unique) value $\mu_\tau$ such that*

$$U_{\mathrm{mar}}(\mu) \leq \tau \iff \mu \geq \mu_\tau.$$

*That is, along the location family $g(\mu)$, thresholding $U_{\mathrm{mar}}$ is order-equivalent to thresholding the mean gap $\mu$.*

*Proof.* Fix $\mu_1 < \mu_2$. For each coordinate $t \in \{1, \ldots, k\}$ we have $\mu_1 + \delta_t < \mu_2 + \delta_t$, and since $\phi$ is strictly decreasing,

$$\phi(\mu_1 + \delta_t) > \phi(\mu_2 + \delta_t).$$

Averaging these $k$ strict inequalities yields

$$\frac{1}{k} \sum_{t=1}^{k} \phi(\mu_1 + \delta_t) > \frac{1}{k} \sum_{t=1}^{k} \phi(\mu_2 + \delta_t),$$

i.e., $U_{\mathrm{mar}}(\mu_1) > U_{\mathrm{mar}}(\mu_2)$. Thus $U_{\mathrm{mar}}$ is strictly decreasing in $\mu$.

Strict monotonicity implies that for any $\tau \in \mathbb{R}$ the equation $U_{\mathrm{mar}}(\mu) = \tau$ has at most one solution; when a solution exists, denote it by $\mu_\tau$. Because $U_{\mathrm{mar}}$ is strictly decreasing, we have

$$U_{\mathrm{mar}}(\mu) \leq \tau \iff \mu \geq \mu_\tau,$$

which shows that thresholding $U_{\mathrm{mar}}$ is equivalent to thresholding $\mu$ along the family $g(\mu)$. $\qquad\square$

*Remark* (Scope). The lemma establishes order-equivalence *only along location shifts* $g(\mu) = \mu \mathbf{1} + \delta$ with fixed shape $\delta$. For general changes of the per-step gap shape (i.e., changing $\delta$), $U_{\mathrm{mar}}$ is still coordinatewise decreasing in each argument, but it is not, in general, a function of the mean alone.

## A.2 Proof of Lemma 2 (boundedness for the small-$N$ variance gate)

Suppose we draw $N$ short stochastic continuations; at step $t$ let $\hat{p}_t$ be the empirical token distribution and define

$$d_t \triangleq 1 - \max_j \hat{p}_t(j) \in [0, 1], \qquad U_{\mathrm{var}}(k, N) \triangleq \frac{1}{k} \sum_{t=1}^{k} d_t.$$

**Lemma 2.** *For every $t$, $\max_j \hat{p}_t(j) \geq \frac{1}{N}$; hence*

$$0 \leq d_t \leq 1 - \frac{1}{N} = \frac{N-1}{N} \implies 0 \leq U_{\mathrm{var}}(k, N) \leq \frac{N-1}{N}.$$

*Moreover, the upper bound is tight when all $N$ samples at a step are distinct.*

*Proof.* Among $N$ samples at step $t$, the modal token appears at least once, so $\max_j \hat{p}_t(j) \geq \frac{1}{N}$ and $d_t \leq 1 - \frac{1}{N}$. Averaging preserves bounds. Tightness: if all samples are distinct then $\max_j \hat{p}_t(j) = \frac{1}{N}$ and $d_t = \frac{N-1}{N}$. $\qquad\square$

A.3    FORMALIZING THE INTUITION IN §3.3: ACCURACY AND BUDGET CALIBRATION

Let $A^{(0)}(q), A^{(1)}(q) \in [0,1]$ denote (calibrated) correctness without/with retrieval for query $q$, and let $\Delta(q) \triangleq A^{(1)}(q) - A^{(0)}(q) \in [-1,1]$. Given a score $U(q)$ and threshold $\tau$, define the retrieval indicator $R_\tau(q) = \mathbf{1}\{U(q) > \tau\}$. The gated accuracy is

$$A_{\text{gate}}(\tau; q) \;=\; A^{(0)}(q) \;+\; \Delta(q)\, R_\tau(q).$$

**Dominance over NEVER-RAG under usefulness calibration.**   Assume there exists $\tau^\star$ s.t.

$$\mathbb{E}[\Delta(q) \mid U(q) \le \tau^\star] \le 0, \qquad \mathbb{E}[\Delta(q) \mid U(q) > \tau^\star] \ge 0.$$

**Proposition 1** (Weak dominance over NEVER). *With $\tau = \tau^\star$,*

$$\mathbb{E}\big[A_{\text{gate}}(\tau)\big] \;\ge\; \mathbb{E}\big[A^{(0)}\big].$$

*Proof.* By the tower rule,

$$\mathbb{E}\big[A_{\text{gate}}(\tau^\star)\big] = \mathbb{E}\big[A^{(0)}\big] + \mathbb{E}\big[\Delta\, R_{\tau^\star}\big] = \mathbb{E}\big[A^{(0)}\big] + \Pr(U > \tau^\star)\, \mathbb{E}[\Delta \mid U > \tau^\star] \;\ge\; \mathbb{E}\big[A^{(0)}\big].$$

$\square$

**When the gate also beats ALWAYS-RAG.**   Since $\mathbb{E}[A^{(1)}] = \mathbb{E}[A^{(0)}] + \mathbb{E}[\Delta]$, we have

$$\mathbb{E}\big[A_{\text{gate}}(\tau^\star)\big] - \mathbb{E}\big[A^{(1)}\big] = \mathbb{E}\big[\Delta\, R_{\tau^\star}\big] - \mathbb{E}[\Delta] = -\mathbb{E}\big[\Delta\, \mathbf{1}\{U \le \tau^\star\}\big].$$

**Proposition 2** (Dominance over ALWAYS under one-sided sign). *If $\Delta(q) \le 0$ almost surely on $\{U(q) \le \tau^\star\}$ (retrieval never helps in the low-U region), then*

$$\mathbb{E}\big[A_{\text{gate}}(\tau^\star)\big] \;\ge\; \mathbb{E}\big[A^{(1)}\big].$$

*Proof.* Under the stated sign condition, $-\mathbb{E}[\Delta\, \mathbf{1}\{U \le \tau^\star\}] \ge 0$.    $\square$

**Budget calibration consistency.**   Let $F_U$ be the CDF of $U$. For a target retrieval rate $\rho \in [0,1]$, set $\tau_\rho = F_U^{-1}(1 - \rho)$. On an i.i.d. development set, the empirical quantile $\hat{\tau}_\rho$ satisfies $\hat{\tau}_\rho \to \tau_\rho$ almost surely, and the realized retrieval rate $\hat{\pi}(\hat{\tau}_\rho) \to \rho$. Thus quantile-based thresholding provides a statistically consistent knob for meeting latency/compute budgets.

**Cost/Lateness decomposition.**   Let $T_{\text{draft}} = k$ be the prefix tokens, $T_{\text{ctx}}$ the (bounded) retrieved context size, and $T_{\text{out}}^{(0)}, T_{\text{out}}^{(1)}$ the output lengths without/with retrieval. With $\pi(\tau) = \Pr(U > \tau)$,

$$\mathbb{E}\big[T(\tau)\big] = T_{\text{draft}} + \big(1 - \pi(\tau)\big)\, \mathbb{E}[T_{\text{out}}^{(0)}] + \pi(\tau)\, \Big(T_{\text{ctx}} + \mathbb{E}[T_{\text{out}}^{(1)}]\Big),$$

so the incremental overhead relative to NEVER is directly governed by $\pi(\tau)$, which is calibrated by the score quantile.

A.4    ABLATION STUDY: RETRIEVER SENSITIVITY (E5-BASE-V2 VS. BGE-M3)

We hold the generator, prompts, corpus, chunking, FAISS type, and decoding fixed, and swap the frozen dual encoder from E5-BASE-V2 to BGE-M3. For each dataset we report NEVER, ALWAYS, and three training-free gates (ENTROPY, MARGIN, VARIANCE). $\Delta$ Latency is added seconds/query over the dataset's NEVER baseline for the same backbone. Thresholds are the representative settings from the main results. For simplicity, the RR in the following context is short for the Retrieve rate, and $\Delta$ stands for the $\Delta$ latency.

Based on Table S2 and S1, we observed three consistent phenomena:

- **Unconditional retrieval remains a poor default, irrespective of retriever.**   Across datasets and both backbones, ALWAYS pays a clear latency tax and frequently underperforms NEVER. With Qwen on TriviaQA, moving from E5 to BGE *worsens* ALWAYS (F1: $57.2 \to 51.7$; $\Delta$ latency: $+3.462 \to +4.663$), highlighting that the bottleneck is retrieval precision rather than the gating policy. Similar patterns appear on PopQA and NQ-Open.

- **The ordering of training-free gates is retriever-agnostic; costs remain minimal.** Under both E5 and BGE, ENTROPY tends to fire more often (e.g., Llama/TriviaQA RR = 0.524), raising overhead with only moderate gains. In contrast, MARGIN and VARIANCE achieve near-best EM/F1 at *tiny* retrieval rates and near-baseline $\Delta$ latency. With Llama/NQ-Open, MARGIN (E5) attains 57.6/54.7 EM/F1 at RR = 0.008 and $\Delta = +0.012$ s, while VARIANCE (BGE) reaches the same 57.6/54.7 EM/F1 at RR = 0.054 and $\Delta = +0.122$ s. On Qwen/TriviaQA, VARIANCE delivers the best BGE quality at 7.4% RR and only $+0.151$ s overhead, mirroring the E5 story.

- **Absolute scores shift idiosyncratically with the retriever, but the *frontier* stays the same.** Entropy with BGE on NQ-Open (Qwen) reaches a slightly higher EM/F1 than with E5 (40.9/39.5 vs. 39.6/39.1) but at a much higher budget (Retrieve rate 0.232 vs. 0.046), while MARGIN/VARIANCE preserve the "near-NEVER latency, better-than-ALWAYS accuracy" property across retrievers. These paired columns make clear that our improvements stem from *when*-to-retrieve decisions, not from retriever-specific quirks.

Table S1: **Retriever sensitivity with Qwen2.5-7B-Instruct.** Each row reports a method; columns pair **E5-base-v2** vs. **BGE-m3**. Trends are consistent across retrievers: MARGIN/VARIANCE improve the accuracy–efficiency frontier with minimal $\Delta$ latency, while ALWAYS pays a latency tax and often underperforms NEVER.

| Dataset | Method | E5-base-v2 | | | BGE-m3 | | |
|---|---|---|---|---|---|---|---|
| | | EM/F1 | RR | $\Delta$(s) | EM/F1 | RR | $\Delta$(s) |
| TriviaQA | NEVER | 60.8/61.4 | 0.000 | 2.947 *(abs.)* | 60.8/61.4 | 0.000 | 2.947 *(abs.)* |
| | ALWAYS | 57.6/57.2 | 1.000 | +3.462 | 52.0/51.7 | 1.000 | +4.663 |
| | ENTROPY | 61.8/62.2 | 0.028 | +0.876 | 62.0/62.6 | 0.028 | +0.404 |
| | MARGIN | **62.2/62.6** | 0.338 | +2.174 | 61.8/62.5 | 0.001 | +0.208 |
| | VARIANCE | 61.8/62.2 | 0.006 | **+0.133** | **62.0/62.7** | 0.074 | **+0.151** |
| PopQA | NEVER | 20.0/20.1 | 0.000 | 2.129 *(abs.)* | 20.0/20.1 | 0.000 | 2.129 *(abs.)* |
| | ALWAYS | 14.6/14.6 | 1.000 | +3.828 | 16.8/16.8 | 1.000 | +4.650 |
| | ENTROPY | 22.4/22.3 | 0.124 | **+1.761** | 22.0/22.1 | 0.418 | +2.776 |
| | MARGIN | **23.0/23.1** | 0.124 | **+1.761** | 22.6/22.5 | 0.002 | **+1.078** |
| | VARIANCE | 22.8/22.9 | 0.182 | +1.847 | **23.0/23.1** | 0.040 | +1.520 |
| NQ-Open | NEVER | 38.8/37.7 | 0.000 | 3.293 *(abs.)* | 38.8/37.7 | 0.000 | 3.293 *(abs.)* |
| | ALWAYS | 37.4/36.7 | 1.000 | +2.922 | 38.2/37.7 | 1.000 | +4.104 |
| | ENTROPY | **39.6/39.1** | 0.046 | +0.964 | **40.9/39.5** | 0.232 | +1.787 |
| | MARGIN | **39.6**/38.8 | 0.304 | +1.295 | 39.0/38.3 | 0.062 | +0.713 |
| | VARIANCE | 38.6/38.0 | 0.012 | **+0.291** | 39.4/37.7 | 0.004 | **+0.121** |

From Table S2 and Table S1, we conclude that the deltas for ALWAYS are diagnostic: whenever retrieval precision dips (e.g., BGE on Qwen/TriviaQA), ALWAYS suffers the most, confirming that indiscriminate context is risky. Second, the behavior of ENTROPY under stronger backbones remains consistent across retrievers: peaked next-token distributions compress entropy and reduce ranking power, so entropy-based gates over-trigger (high RR) and add latency without commensurate gains. Third, MARGIN and VARIANCE retain discriminative range because the top-1/top-2 gap and small-$N$ disagreement track *instability* in the prefix; that instability correlates with cases where external evidence flips or stabilizes the answer. Finally, reading efficiency through $\Delta$ latency shows the budget clarity of gating: the best MARGIN/VARIANCE points cluster near the NEVER floor while outperforming ALWAYS, *for both retrievers*.

In short, the TARG method is **retriever-agnostic**. Upgrading the retriever raises absolute ceilings for *all* modes, but a calibrated MARGIN/VARIANCE gate continues to avoid wasteful retrieval on easy inputs and to dominate ALWAYS on the accuracy–efficiency frontier. This directly supports the claim that training-free, budget-aware gating is a portable primitive for RAG systems.

Table S2: **Retriever sensitivity with Llama-3.1-8B-Instruct.** Same layout as Table S1. With a stronger backbone, MARGIN/VARIANCE reach near-best quality at vanishing retrieval rates and near-zero Δ latency under *both* retrievers, while ENTROPY over-triggers.

| Dataset | Method | E5-base-v2 | | | BGE-m3 | | |
|---|---|---|---|---|---|---|---|
| | | **EM/F1** | **RR** | **Δ(s)** | **EM/F1** | **RR** | **Δ(s)** |
| TriviaQA | NEVER | 80.8/80.0 | 0.000 | 10.383 *(abs.)* | 80.8/80.0 | 0.000 | 10.383 *(abs.)* |
| | ALWAYS | 67.6/67.2 | 1.000 | +1.069 | 66.8/65.4 | 1.000 | +1.581 |
| | ENTROPY | 74.4/74.1 | 0.524 | +0.495 | 75.6/74.9 | 0.524 | +0.435 |
| | MARGIN | **83.8/83.0** | 0.001 | **+0.018** | **83.6/82.9** | 0.002 | +0.034 |
| | VARIANCE | 83.6/**83.0** | 0.001 | **+0.018** | 83.4/82.6 | 0.001 | **+0.025** |
| PopQA | NEVER | 35.2/34.4 | 0.000 | 10.299 *(abs.)* | 35.2/34.4 | 0.000 | 10.299 *(abs.)* |
| | ALWAYS | 24.8/24.6 | 1.000 | +1.269 | 26.0/25.8 | 1.000 | +0.700 |
| | ENTROPY | 28.8/28.8 | 0.760 | +0.974 | 32.4/32.4 | 0.760 | +0.497 |
| | MARGIN | **36.6/36.2** | 0.108 | +0.424 | 36.2/35.6 | 0.001 | **+0.063** |
| | VARIANCE | 36.4/36.0 | 0.084 | **+0.317** | **36.2/35.8** | 0.028 | +0.290 |
| NQ-Open | NEVER | 53.8/51.7 | 0.000 | 10.299 *(abs.)* | 53.8/51.7 | 0.000 | 10.299 *(abs.)* |
| | ALWAYS | 48.6/46.1 | 1.000 | +1.248 | 46.2/44.8 | 1.000 | +1.670 |
| | ENTROPY | 55.4/53.1 | 0.132 | +0.175 | 53.8/51.3 | 0.228 | +0.505 |
| | MARGIN | **57.6/54.7** | 0.008 | **+0.012** | 57.0/54.1 | 0.008 | **+0.028** |
| | VARIANCE | 56.8/53.7 | 0.026 | +0.059 | **57.6/54.7** | 0.054 | +0.122 |

Table S3: Prefix-length ablation for TARG (gate fixed). Each cell shows **EM / F1 [Retrieval Rate]** in %. Bold indicates the best EM (primary) at each threshold. A 20-token draft (k=20) yields the best overall quality at mid thresholds while avoiding the high retrieval of longer prefixes.

| Prefix $k$ | $\tau=0.3$ | $\tau=0.5$ | $\tau=0.65$ | $\tau=0.8$ |
|---|---|---|---|---|
| $k=10$ | 38.6 / 37.5 [32.6] | 39.2 / 37.7 [11.0] | 39.2 / 38.2 [6.0] | 39.4 / 38.3 [2.8] |
| $k=20$ | 38.2 / 37.2 [56.0] | **40.8 / 39.4** [23.2] | 39.8 / **39.2** [13.0] | **39.8** / **38.8** [6.0] |
| $k=30$ | **39.6 / 38.5** [66.0] | 40.0 / 38.5 [31.2] | **40.0** / 38.7 [16.0] | 39.0 / 37.7 [7.2] |

## A.5  ABLATION STUDY: PREFIX LENGTH $k$.

Table S3 indicates that mid thresholds favor a 20-token prefix. At $\tau=0.5$, $k=20$ attains the best overall quality (EM 40.8, F1 39.4) at a moderate retrieval rate (23.2%), which is also the global optimum in the sweep—suggesting 20 tokens are sufficient to stabilize the prefix-uncertainty signal without over-triggering retrieval. Longer prefixes raise the retrieval rate without consistent gains: at $\tau=0.3$, $k=30$ improves EM/F1 by only 1-1.3 points yet drives retrieval to 66% (vs. 56% for $k=20$ and 32.6% for $k=10$), an unfavorable quality–cost trade. Shorter prefixes under-inform the gate: with $k=10$ at $\tau=0.5$, quality lags (EM 39.2 / F1 37.7) despite frugal retrieval (11.0%), implying ten tokens often fail to expose the uncertainty patterns that predict retrieval benefit. At higher thresholds ($\tau=0.65$–0.8), quality differences narrow, but $k=20$ remains best or tied on EM and best on F1 while keeping retrieval low (6–13%). **Overall, a 20-token draft offers the best quality–cost balance**; we adopt $k=20$ as the default and calibrate the threshold at mid values (e.g., via score quantiles) to meet a target retrieval budget.

## A.6  UNCERTAINTY VS. BASE-MODEL ACCURACY

To empirically examine whether the scalar scores $U$ behave as proxies for parametric knowledge, we analyze *base-model* accuracy (Never-RAG) as a function of uncertainty. For each dataset and gate:

1. We run the zero-RAG baseline and compute the gate score $U(q)$ from the $k$-token prefix for every query $q$.

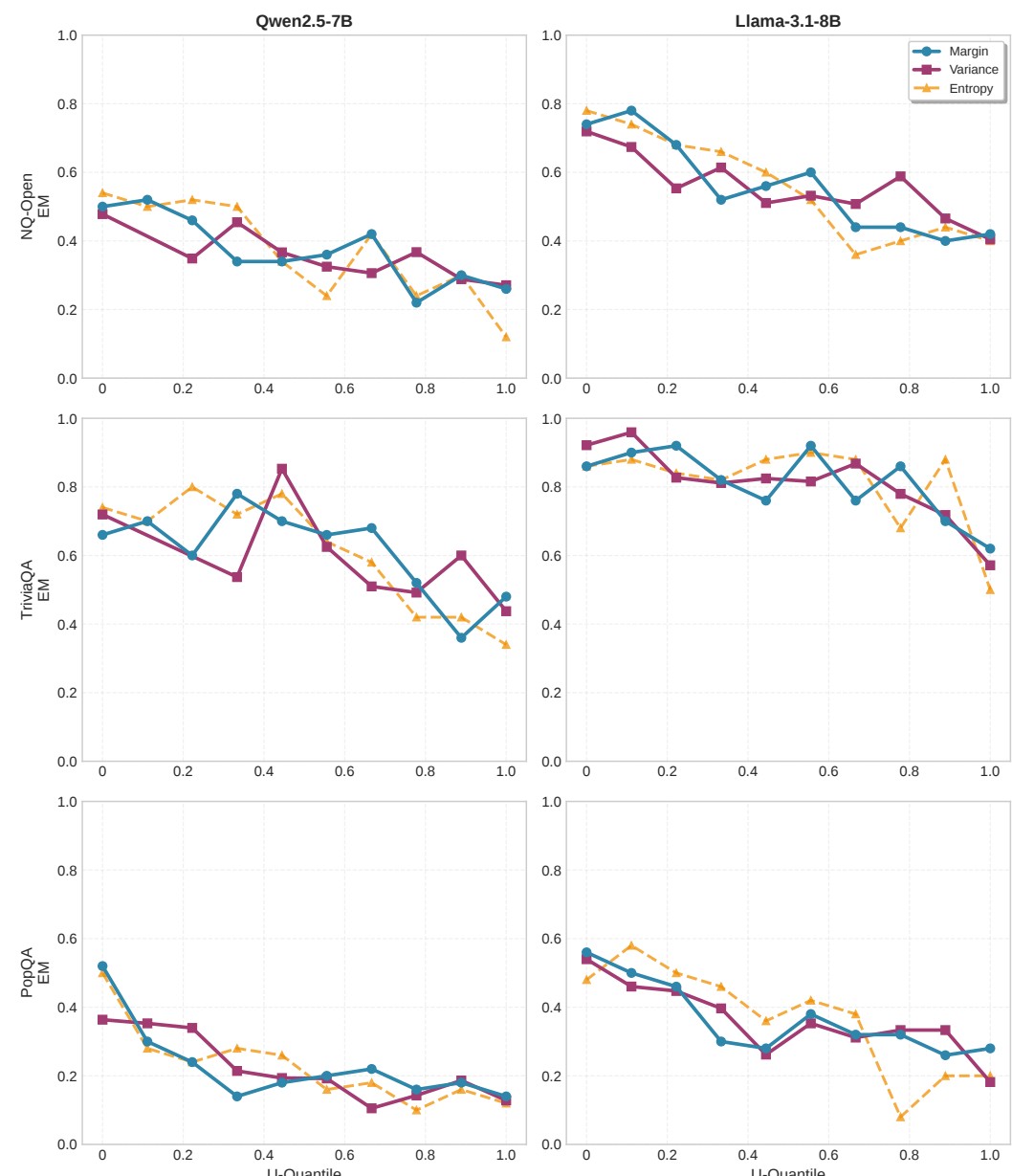

Figure S1: **Base-model accuracy vs. uncertainty quantiles.** For each gate and dataset, we bin queries by the score $U(q)$ and plot Never-RAG EM in each bin. Margin and variance gates show a clear decreasing trend, supporting their use as training-free proxies for parametric uncertainty.

2. We partition queries into $B$ bins (we use deciles, $B=10$) according to $U(q)$, from lowest to highest uncertainty.

3. Within each bin, we compute EM for the base generator under Never-RAG.

Figure S1 reports EM as a function of the $U$-quantile for Qwen2.5-7B and Llama-3.1-8B on NQ-Open, TriviaQA, and PopQA. For the MARGIN and VARIANCE gates, base-model accuracy decreases monotonically (or near-monotonically) as $U$ increases on all datasets, indicating that these scores are informative proxies for epistemic uncertainty about parametric knowledge. The trend is strongest under Llama-3.1-8B, where the distribution is sharper and logit gaps/ensemble disagreement retain more discriminative power. The ENTROPY score exhibits a weaker trend, particularly

for Llama-3.1-8B, consistent with our main results: prefix entropies compress under sharper models and lose ranking power relative to margin- or variance-based signals.

### A.7 ERROR QUADRANTS OVER $(U(q), \Delta(q))$

We next characterize when retrieval helps or hurts by analyzing quadrants over the uncertainty score $U(q)$ and the retrieval benefit $\Delta(q)$ introduced in Section 3.3. For each query $q$ in a development split, we compute:

- $A^{(0)}(q)$ and $A^{(1)}(q)$: correctness under Never-RAG and Always-RAG;
- $\Delta(q) \in \{-1, 0, 1\}$, defined as $+1$ if Always-RAG is correct and Never-RAG is wrong, $-1$ if the converse holds, and $0$ otherwise;
- the margin-gate score $U(q)$ from the $k$-token prefix (same $k$ as in the main experiments).

We then fix the same threshold $\tau$ used in our main TARG runs and define *low* vs. *high* uncertainty by $U(q) \leq \tau$ and $U(q) > \tau$, respectively. This yields four quadrants:

(a) Low $U$, $\Delta > 0$: base model is wrong and retrieval would help ("confidently wrong, retrieval useful");

(b) High $U$, $\Delta < 0$: model is uncertain and retrieval is harmful ("uncertain, retrieval harmful");

(c) High $U$, $\Delta > 0$: model is uncertain and retrieval is beneficial (ideal region for TARG to retrieve);

(d) Low $U$, $\Delta \leq 0$: base model is safe to run without retrieval (retrieval neutral or harmful).

Table S4: **Quadrant analysis over** $(U(q), \Delta(q))$ on a dev split, using the margin gate and the same $\tau$ as in our main experiments. Values are percentages of queries in each quadrant (rows sum to 100%). Model: **Qwen2.5-7B-Instruct**.

| Dataset | (a) Low $U$, $\Delta > 0$ | (b) High $U$, $\Delta < 0$ | (c) High $U$, $\Delta > 0$ | (d) Low $U$, $\Delta \leq 0$ |
|---|---|---|---|---|
| NQ-Open | 6.0% | 2.2% | 4.0% | 87.8% |
| TriviaQA | 5.4% | 3.8% | 2.0% | 88.8% |
| PopQA | 1.2% | 9.6% | 7.2% | 82.0% |

Table S5: **Quadrant analysis over** $(U(q), \Delta(q))$ on a dev split, using the margin gate and the same $\tau$ as in our main experiments. Values are percentages of queries in each quadrant (rows sum to 100%). Model: **Llama-3.1-8B-Instruct**.

| Dataset | (a) Low $U$, $\Delta > 0$ | (b) High $U$, $\Delta < 0$ | (c) High $U$, $\Delta > 0$ | (d) Low $U$, $\Delta \leq 0$ |
|---|---|---|---|---|
| NQ-Open | 1.4% | 10.2% | 5.8% | 82.6% |
| TriviaQA | 0.6% | 15.2% | 2.6% | 81.6% |
| PopQA | 0.4% | 20.0% | 9.8% | 69.8% |

Tables S4 and S5 report the percentage of dev queries in each quadrant for Qwen2.5-7B-Instruct and Llama-3.1-8B-Instruct, using the margin gate and the same thresholds $\tau$ as in our main experiments.

Several trends are noteworthy. First, across all datasets and both models, the majority of queries lie in quadrant (d): low-uncertainty queries for which retrieval is neutral or harmful (82–89% for Qwen2.5, 70–83% for Llama-3.1). In these cases, TARG's decision to *not* retrieve when $U(q) \leq \tau$ is aligned with the sign of $\Delta(q)$ and avoids unnecessary context.

Second, the problematic quadrant (a)—low $U$ but $\Delta > 0$, where the gate would skip retrieval even though it improves accuracy—is relatively small on all datasets, and becomes very small once we move to the stronger Llama-3.1 backbone (e.g., $1.4\%$ on NQ-Open, $0.6\%$ on TriviaQA, $0.4\%$ on PopQA). This indicates that genuinely "confidently wrong, but fixable by retrieval" cases are rare under the margin gate, especially for the sharper model.

Third, the high-uncertainty region concentrates most of the examples where retrieval meaningfully changes outcomes. On Qwen2.5, the combined mass of quadrants (b) and (c) is modest (typically 6–17%), reflecting that Always-RAG differs from Never-RAG on a relatively small set of questions. On Llama-3.1, a larger fraction of queries fall into (b) and (c), particularly on PopQA (29.8% total), with a substantial subset in (c) (e.g., $9.8\%$ for PopQA), where retrieval improves accuracy precisely when the model is uncertain. Quadrant (b)—high $U$ but $\Delta < 0$—captures cases where the retriever surfaces distractors or entity aliases and retrieval becomes harmful; its larger mass on PopQA and TriviaQA highlights that these datasets pose a more difficult retrieval problem.

Overall, this quadrant analysis supports the usefulness-calibration assumption of Section 3.3: while $U(q)$ and $\Delta(q)$ are not perfectly aligned on every query, low-uncertainty queries are overwhelmingly those where retrieval is unnecessary or harmful, and high-uncertainty queries are where retrieval is most likely to change the answer (either positively or negatively). TARG therefore concentrates retrieval in the region where external evidence is most consequential, while keeping the risk of skipping beneficial retrieval (quadrant (a)) small, especially for the stronger backbone.

The quadrant analysis above is descriptive of the benchmark distributions under our specific stack (frozen dense retriever over chunked Wikipedia) and serves to check the usefulness-calibration assumption in Section 3.3 empirically. In particular, we observe that the problematic "low-$U$, $\Delta > 0$" region (quadrant (a)) is small, while high-$U$ mass (quadrants (b)+(c)) concentrates queries where retrieval changes the answer. However, this behavior is not universal. In adversarial or tiered routing settings that deliberately concentrate traffic in quadrants (a) and (b), the conditional expectations in the usefulness-calibration inequalities need not hold, and a single threshold $\tau_*$ need not exist. Such regimes lie outside the scope of our simple dominance argument in Section 3.3 and would require re-calibration of the gate or alternative control strategies under that shifted distribution.

## A.8 SENSITIVITY TO THRESHOLD $\tau$ AND PREFIX LENGTH $k$

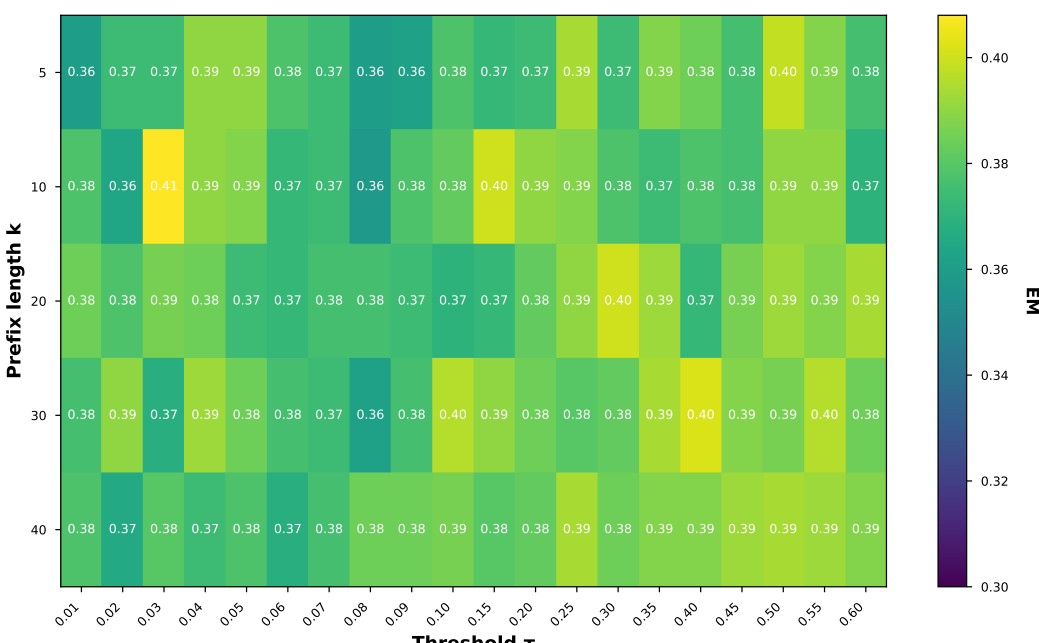

Figure S2: **EM as a function of** $(k, \tau)$ for the Margin gate on NQ-Open, Qwen2.5-7B-Instruct model. A broad region of high performance indicates robustness to the choice of prefix length and threshold.

To assess the cost of calibration, we perform a more extensive sweep over the prefix length $k$ and the decision threshold $\tau$ for the margin gate. On NQ-Open, for each $k \in \{5, 10, 20, 30, 40\}$ we:

1. compute the margin score $U_{\mathrm{mar}}(q)$ for every dev query using the $k$-token prefix, and

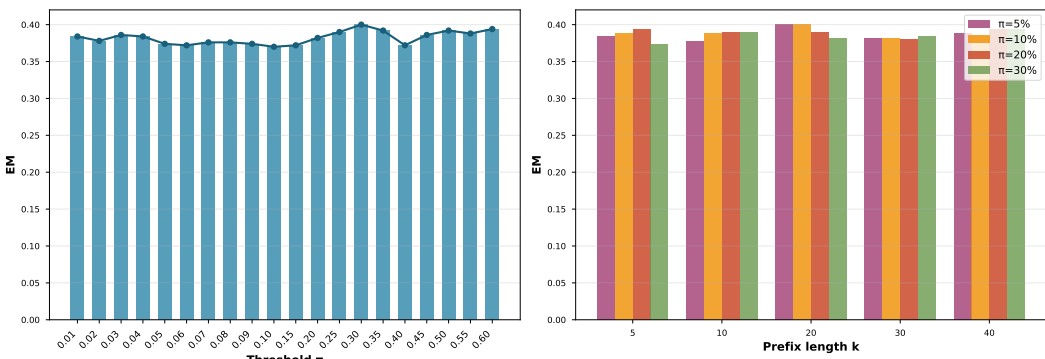

Figure S3: **1D sensitivity slices** for Qwen 2.5-7B-Instruct model on NQ-Open dataset, using margin gate. Left: EM versus $\tau$ for fixed $k=20$. Right: EM versus $k$ when $\tau$ is chosen to match a fixed retrieval budget. Performance varies smoothly, supporting the claim that calibration is cheap and robust.

   2. evaluate EM over a grid of thresholds $\tau$ corresponding to fixed quantiles of the empirical CDF of $U_{\mathrm{mar}}$ (we use 20 equally spaced quantiles).

Figure S2 shows EM as a function of $(k, \tau)$ for Qwen2.5-7B-Instruct and Llama-3.1-8B-Instruct. Two trends emerge:

- There is a broad region of good performance: for each model, a wide band of $(k, \tau)$ values yields EM close to the best observed value, rather than a single needle-like optimum.

- Moderate mis-specification of either $k$ or $\tau$ has limited impact on EM and retrieval rate, especially in the regime of modest retrieval budgets ($\pi$ between $0.05$ and $0.3$).

Figure S3 provides 1D slices: EM versus $\tau$ for fixed $k=20$ and EM versus $k$ when $\tau$ is chosen to match a fixed retrieval budget. These plots confirm that TARG does not require fine-grained tuning and that a simple one-dimensional grid search over $\tau$ on a small dev set suffices in practice.

### A.9 Dynamic re-check ablation

Algorithm 1 admits an online variant in which the gate can be re-evaluated periodically and retrieval can be triggered mid-generation. Concretely, we consider a simple dynamic re-check policy on NQ-Open with Qwen2.5-7B-Instruct and the margin gate: starting from the base prompt $B(q)$, we decode in fixed blocks of $m$ tokens, recompute $U(q)$ on the running prefix after each block, and, if $U(q) > \tau$ at any checkpoint and retrieval has not yet occurred, we retrieve once and continue decoding from $B(q) \oplus C$.

Figure S4 compares this dynamic re-check variant against single-shot TARG as a function of the threshold $\tau$ (from $0.1$ to $0.8$), reporting EM, retrieval rate, and latency overhead relative to the Never-RAG baseline.[2] Several trends emerge:

- For low to moderate thresholds ($\tau \in \{0.1, 0.2, 0.3, 0.4\}$), the dynamic re-check policy is extremely aggressive. For example, at $\tau=0.3$, single-shot TARG achieves EM $= 39.0\%$ with a retrieval rate of $6.2\%$ and $\approx 4.1$ s/query latency, whereas dynamic re-check achieves EM $= 38.8\%$ with a retrieval rate of $88.2\%$ and $\approx 20.6$ s/query latency, i.e., $\sim 16.6$ s extra wall-clock time per query for essentially no accuracy gain. At $\tau=0.1$, retrieval rate saturates at $100\%$ and latency almost doubles relative to single-shot.

- At higher thresholds ($\tau \geq 0.5$), single-shot TARG rarely or never retrieves (retrieval rate $0$–$10\%$), while dynamic re-check continues to incur a large latency overhead ($\approx 16$–$17$ s/query) despite only modest changes in EM (improvements of at most about 2 points).

---

[2]Implementation details and exact numbers are provided in the released scripts.

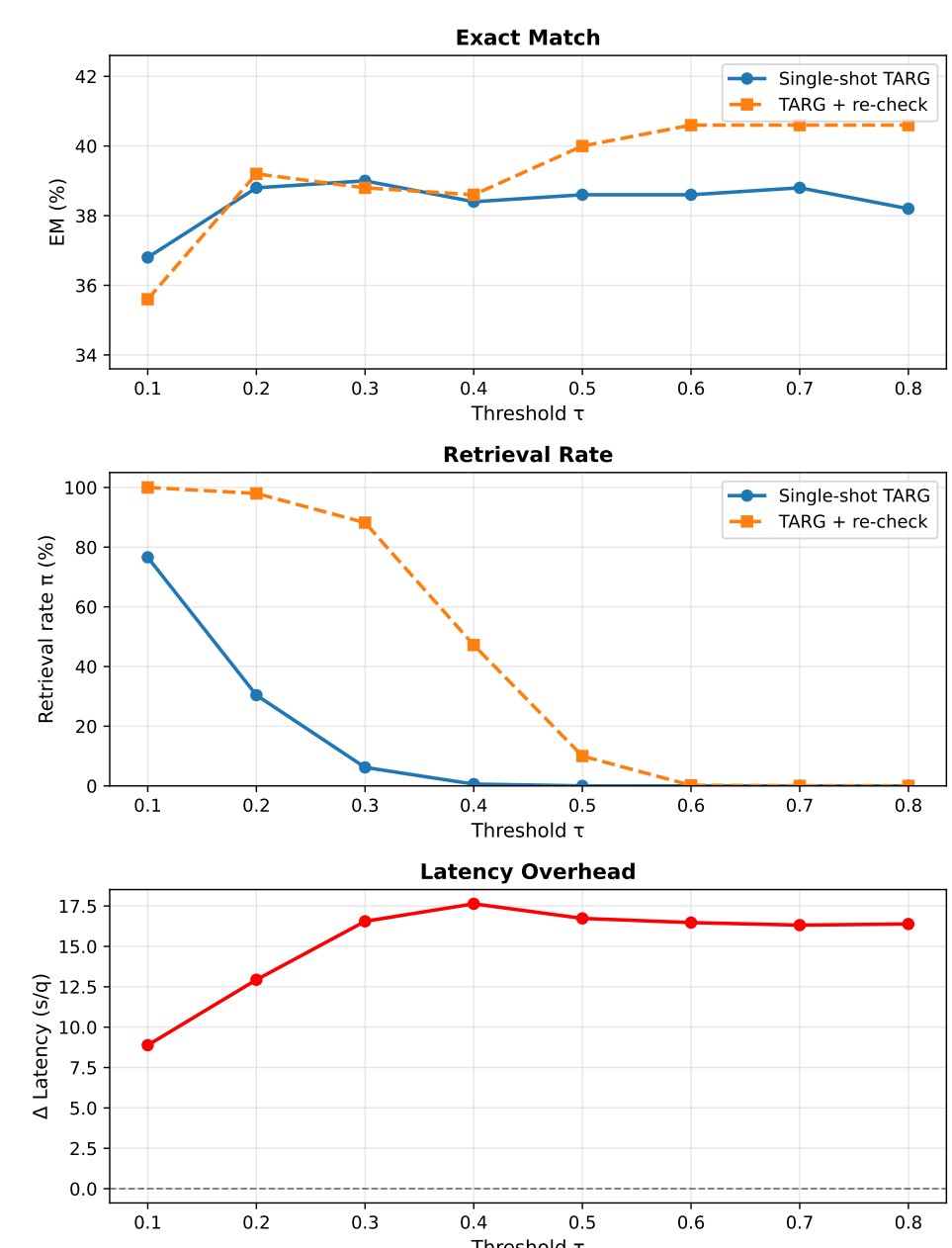

Figure S4: **Dynamic re-check vs. single-shot TARG on NQ-Open (Qwen2.5-7B, Margin gate).** Exact Match (top), retrieval rate (middle), and latency overhead relative to Never-RAG (bottom) as a function of the threshold $\tau$. Dynamic re-check yields similar EM but dramatically higher retrieval and latency at many thresholds, motivating our choice to use single-shot gating in all main experiments.

This reflects the cost of repeatedly scoring prefixes during generation, even when retrieval is rarely triggered.

- Across the entire sweep, EM differences between single-shot and dynamic re-check remain small (within $\approx \pm 2$ points), whereas retrieval rate and latency can increase dramatically under dynamic re-check, especially at thresholds that are otherwise reasonable for single-shot gating.

Overall, this ablation shows that a naive dynamic re-check policy with a fixed threshold $\tau$ is not cost-effective for the short-answer QA regime studied in this work: it either degenerates toward Always-RAG (very high retrieval rates and large latency overhead) or incurs substantial extra computation for marginal accuracy changes. For this reason, we adopt *single-shot* TARG in all main experiments and view dynamic re-check as an extension that may be better suited to long-form generation with more sophisticated, possibly adaptive thresholds.

