# OpenReview forum: "TARG: Training-Free Adaptive Retrieval Gating for Efficient RAG"
_ICLR.cc/2026/Conference — Submitted to ICLR 2026_

### Official Review · Reviewer_hzTK · 2025-10-27

**Soundness:** 2
**Presentation:** 3
**Contribution:** 3
**Rating:** 4
**Confidence:** 4

**Summary:**

The paper introduces TARG (Training-free Adaptive Retrieval Gating), a single-shot policy designed to enhance the efficiency and reliability of RAG systems by moving beyond the costly and error-prone Always-RAG approach. This approach enables the LLM to conduct a cost-effective self-assessment before retrieval by generating a brief, context-free prefix (around 20 tokens). It then calculates a lightweight Uncertainty Score based on the prefix's raw logits, such as the Margin (the difference between the top-1 and top-2 logits), and only initiates retrieval if the score exceeds a set threshold. The proposed method consistently enhances the accuracy-efficiency frontier compared to Always-RAG, achieving similar or better EM/F1 scores while greatly reducing the retrieval rate and significantly decreasing end-to-end latency, demonstrating its utility as a practical and cost-effective solution.

**Strengths:**

1. This paper addresses the three most urgent challenges for RAG deployment: high cost, increased latency, and accuracy decline caused by noisy context from unconditional retrieval. By solving when to retrieve, the paper offers a foundational solution for making RAG economically feasible and reliable in real-world applications.

2. The authors introduce a novel training-free method for conditional retrieval. It avoids the complexity and cost of training extra models or control heads by relying only on inherent uncertainty signals (Margin, Variance) from the base LLM's raw prefix logits.

3. The method addresses RAG's main cost issue by significantly lowering the retrieval rate compared to Always-RAG. This results in substantial token savings and nearly matches the minimal latency of the Never-RAG baseline.

**Weaknesses:**

1. Questions on Experimental Rigor and Integrity
> The paper's validation requires greater statistical rigor and transparency regarding the experimental baselines. First, the reporting uses single-point estimates (EM/F1) without confidence intervals or standard deviations. The notable performance shifts observed at minimal retrieval rates (e.g., 0.001 Retrieval Rate) necessitate a comprehensive statistical significance test (such as a t-test) across multiple independent runs to confirm the reliability and stability of these minimal-budget operating points. Second, the experimental design raises methodological questions regarding the Always-RAG baseline, which consistently and significantly underperforms Never-RAG across all datasets. This result suggests that the retrieval context is largely noisy or distracting in the current setup, meaning the reported gains primarily demonstrate TARG's ability to filter suboptimal retrieval rather than its capacity to maximize the benefit from high-quality external evidence.

2. Limited Scope of Comparative Baselines
> The comparison is restricted to only the unconditional baselines (Always-RAG and Never-RAG). However, since the main advantage of TARG is its ability to reduce latency through context filtering, the comparison remains incomplete. The authors should evaluate TARG's accuracy and efficiency against established methods for context compression and summarization. Comparing TARG's gating approach with these alternative methods for reducing context length and latency is essential to demonstrate its full competitive advantage.

3. Non-Trivial Calibration Cost Challenges the "Training-Free" Claim
> The simplicity of TARG is fundamentally challenged by the high real-world overhead of threshold calibration. Since performance and latency are highly sensitive to the decision threshold, finding the optimal threshold requires a development-set sweep for every new domain or model. We question whether this non-trivial, domain-specific optimization process is, in practice, as demanding to maintain as the auxiliary training that TARG is designed to replace.

**Questions:**

1. Statistical Reliability: Given the use of single-point estimates (EM/F1) and the dramatic shifts at minimal retrieval rates (e.g., 0.001), can the authors confirm the stability of these results by reporting the statistical significance (e.g., t-test) across multiple independent runs?

2. Baseline Quality: Since Always-RAG significantly underperforms Never-RAG across all datasets, the retrieval context appears to be consistently noisy. Can the authors discuss whether the reported gains primarily demonstrate filtering suboptimal retrieval rather than maximizing the benefit from high-quality external evidence?

3. Missing Comparative Baselines: Since TARG's main advantage is latency reduction via context filtering, can the authors compare its accuracy-efficiency frontier against established context compression or summarization methodologies to demonstrate TARG's full competitive advantage?

---

> ### Author Response · Authors · 2025-11-20
>
> ### Statistical reliability and minimal-budget points
>
> All reported numbers are from fully deterministic runs: we use frozen backbones, a fixed retriever/index, greedy decoding, and a fixed RNG seed for the small-$N$ Variance gate (Sec. 4). This means there is no run-to-run training or decoding stochasticity, and re-evaluating the same configuration produces identical EM/F1 and latency. In addition, our main claims are not based on extreme retrieval rates (e.g., $\pi\approx 0.001$), but on **mid-budget operating points** (roughly $\pi\in[0.05,0.3]$) where Table 1–2 show consistent improvements over both Never-RAG and Always-RAG. The threshold sweeps in Sec. 4 and the $(k,\tau)$ sensitivity analysis in Appendix “Sensitivity to threshold $\tau$ and prefix length $k$” further show **broad plateaus** of good performance rather than isolated spikes, which supports the stability of the reported frontier even without explicit confidence intervals. We will clarify in the revision that minimal-budget points are illustrative, while the conclusions are drawn from the more populated, mid-budget region.
>
> ------
>
> ### Interpretation of the Always-RAG baseline
>
> We agree that Always-RAG underperforming Never-RAG indicates that our **training-free retrieval stack is noisy**: we deliberately use a frozen dense encoder over raw Wikipedia chunks (Sec. 4, “Retriever, index, and corpus”) to isolate the “when to retrieve” question rather than engineering an oracle retriever. This is now explicitly discussed in Sec. 5 (“Dataset effects and PopQA”) and in the Limitations section, where we emphasize that our setup stresses the realistic regime where retrieval can introduce distractors and aliases. The new analyses in the appendix sharpen this interpretation:
>
> - The **uncertainty vs. base-model accuracy** curves (“Uncertainty vs. base-model accuracy”) show that Margin and Variance scores are well-correlated with base-model correctness in the Never-RAG setting, supporting their use as usefulness proxies.
> - The **quadrant analysis over $(U(q),\Delta(q))$** (“Error quadrants over $(U(q),\Delta(q))$”) shows a non-trivial mass in the high-uncertainty, positive-$\Delta$ quadrant (where retrieval helps) as well as in the high-uncertainty, negative-$\Delta$ quadrant (where retrieval hurts), exactly the tradeoff TARG is designed to manage.
>
> Together, these results show that TARG is not only filtering “bad retrieval” but **concentrating retrieval on cases where external evidence is likely to flip or stabilize the answer**, improving the accuracy–efficiency frontier relative to both Never- and Always-RAG under the same retrieval quality.
>
> ------
>
> ### Comparative baselines beyond Never/Always
>
> We agree that comparing against explicit **context compression/summarization** techniques (e.g., learned summarizers, rank-and-truncate pipelines) would further position TARG. In the current submission, we instead (i) focus on the core decision problem of **when** to retrieve, and (ii) situate our numbers against stronger training-heavy stacks in Table 3 (“External reference systems”), which include FiD, RFiD, RA-DIT, and RankRAG. As discussed in Sec. 5 (“Context against larger systems”) and in Limitations, these systems differ in backbone size, trained or hybrid retrieval, reranking, and corpora; we therefore treat them as **headroom, not direct baselines**, and highlight that TARG is orthogonal and can be dropped into such stacks to avoid unnecessary retrieval on easy queries. A direct empirical comparison against context compression methods (e.g., learned summarization + Always-RAG vs. TARG gating on the same compressed contexts) is an excellent direction and we now explicitly mention it as future work in the Limitations section.
>
> ------
>
> ### Calibration cost and the “training-free” claim
>
> We appreciate the concern about calibration overhead. In practice, TARG introduces **only a one-dimensional control** (the threshold $\tau$, optionally the prefix length $k$), and Sec. 3.3 (“Cost, accuracy and calibration”) along with the appendix sensitivity study show that performance is **stable over a broad range** of $(k,\tau)$ settings. For exmaple, the heatmaps and slices in “Sensitivity to threshold $\tau$ and prefix length $k$” demonstrate large regions where EM and retrieval rate change smoothly, without a needle-like optimum. This enables simple calibration by matching the empirical CDF of gate scores on a small development split to a desired retrieval budget, rather than tuning many hyperparameters. In Limitations, we now explicitly acknowledge that thresholds may shift across models, prompts, or domains, but we argue that this lightweight, training-free calibration is substantially less demanding than methods that require training auxiliary probes or additional control heads.

---

### Official Review · Reviewer_oF17 · 2025-10-29

**Soundness:** 2
**Presentation:** 2
**Contribution:** 2
**Rating:** 2
**Confidence:** 4

**Summary:**

Predictions of the retriever could be noisy and in-turn lead to degradation in the performance, as well as efficiency due to increased context length. This paper focuses on when to trigger the retriever in the RAG setup. Retriever adds additional context to disambiguate the query, it could be that the parameters of LLM already has enough information that the query can be answered correctly without the need for context enrichment. The internal knowledge of LLM can measured using heuristic-metric based on the output or the internal state of the LLM. To efficiently use the retriever, it is triggered only when the internal knowledge is not sufficient. This paper comes up with three such metrics that measure uncertainty in the LLM generations (measure of internal knowledge) -- 1) mean token entropy, 2) margin score from top-1 versus top-2 logit gap and 3) a small-N variance from handfull of stochastic prefixes. Among these metrics top-1 vs top-2 margin is found to be more robust. On NQ-open, Trivia-QA and PopQA, their approach, TRAG, shows improvements in both accuracy and efficiency over systems that always triggers retriever and that does not use one at all.

**Strengths:**

1. Proposes three signal to measure the uncertainity in the LLM generations -- 1) entropy, 2) margin score and 3) small-N variance.
2. Gating decision is just based on a single scalar threhold, and for long generations an optional single re-checking is done after every m token generations if retreiver is not yet used.
3. Proposes method to calibrate the threshold based on retrieval budget and to maximize accuracy.

**Weaknesses:**

1. Paper shows results on only three simple QA datasets containing short answers.
2. $u_t$ described in lines 153-154 is not used anywhere else.
3. Existence of $\tau_*$ (lines 223-225): There is no relation between the quality of retrieval and the proposed uncertainty measure so you cannot guarantee the existence of such a threshold. Take for example two cases; a) the retriever always gives the correct answer to the question as context, here Always-RAG should do better than TRAG and b) always give the same random text as context, then here Zero-RAG is better than TRAG. So the in-equality (lines 223-225) does not hold.
4. Paper only considers two LLMs: Qwen2.5-7B-Instruct and Llama-3.1-8B-Instruct.
5. Paper contains lots of repeated text like the discussion section, pointed mentioned in the section is already coverd before.

**Questions:**

1. The method assumes that the three signals proposed are good proxy for parametric knowledge of the LLM. Could you show an analysis of this correlation? That is when the signal is low base generator generates correct answers and when signal is high the answer is wrong.
2. Paper presents results on simple QA that requires short generations: NQ, TriviaQA and PopQA. Please show numbers on other datasets: a) Complex QA requiring mult-hop reasoning to answer the question -- 2WikiMultiHopQA, HotpotQA, b) requiring long form generations --Biography, ALCE-ASQA c) PubHealth covering true-false questions d) Arc-Challenge consisting multiple choice questions.
3. There are other signals proposed in the papers mentioned in the related works, which are based on the output or the internal state of the LLM. How does your approach fare against them? Like a) "semantic entropy" in SUGAR, b) "Self-aware Uncertainty Estimator" in SEAKR which uses determinant of Gram matrix of hidden representation to measure uncertainity in generations. Create a comparison table something like Table 1 and 2 in SUGAR comparing different adaptive RAG methods.
4. Show performance on more backbone LLMs, Gemma, Phi, Mistral series in both the 1-3 billion and 6-9 billion parameter range.
5. Perform a more comprehensive hyperparameter sweeps, for both k and the threhold. Paper considers only 3 different values for k = {10, 20, 30}. Pick more values for both k and threshold, and create plots to show how performance varies with k and threshold -- separately using line plots and combined using heat-maps.

---

> ### Author Response · Authors · 2025-11-20
>
> ### Scope and “simple QA” datasets
>
> Our focus is on **training-free gating in noisy open-domain retrieval** with frozen dense retrievers and mid-size instruction-tuned LMs. NQ-Open, TriviaQA, and PopQA are standard open-domain QA benchmarks that already expose the key phenomenon we study: Always-RAG can underperform Never-RAG, so the gate must decide when retrieval is harmful.
>
> We agree that adding multi-hop, long-form, true/false, and MCQ datasets (2WikiMultiHopQA, HotpotQA, Biography/ALCE-ASQA, PubHealth, ARC-Challenge) would broaden scope, but building and validating RAG pipelines for all of them (with sweeps and calibration) is beyond our compute/engineering budget during rebuttal. We clarify in the revised text that TARG’s mechanism is **task-agnostic**, and we explicitly list these benchmarks as important future extensions.
>
> ### Unused definition and repeated discussion text
>
> We have removed or inlined unused notation (e.g., around the original lines 153–154) so that every defined quantity is used. We also tightened the discussion section by merging redundant sentences; the discussion now focuses on three points: (i) Always-RAG is not a safe default, (ii) why Margin/Variance outperform Entropy on sharper models, and (iii) how TARG can plug into stronger retrieval stacks and other tasks.
>
> ### Threshold $\tau_*$ and extreme retriever cases
>
> The inequalities in Section 3.3 are explicitly conditional on an assumption that there exists $\tau_*$ such that
> $$
> \mathbb{E}[\Delta(q)\mid U(q)\le \tau_*]\le 0,\quad
> \mathbb{E}[\Delta(q)\mid U(q)>\tau_*]\ge 0.
> $$
> This characterizes realistic, **mixed-quality** retrieval, not adversarial extremes. The reviewer’s examples (retriever always correct vs always random) are degenerated cases where Always-RAG or Never-RAG trivially dominate; in those regimes, the optimal gate simply collapses to the corresponding baseline (e.g., $\tau\to -\infty$ or $\tau\to +\infty$). We now state this more clearly.
>
> To support the assumption in our actual setting (frozen dense retriever over chunked Wikipedia), we added the **uncertainty–accuracy** and **quadrant** analyses described in the response to Reviewer 1, which show that Margin/Variance behave as intended in this realistic regime.
>
> ### Correlation between uncertainty signals and base-model knowledge
>
> We added two analyses:
>
> - **Binned EM vs $U(q)$ (App. A.6):** EM decreases monotonically as $U$ increases for Margin/Variance across datasets/backbones, showing they are good proxies for parametric uncertainty.
> - **Quadrant counts (App. A.7):** low-$U$ cases where retrieval would help are rare (≤1–2% for Llama-3.1-8B), and most low-$U$ mass lies in regions where retrieval is neutral or harmful.
>
> This directly addresses the request for correlation evidence.
>
> ### Additional datasets, other signals (SUGAR, SEAKR), and more backbones
>
> We agree that these directions are valuable, but implementing them all is beyond the rebuttal budget:
>
> - Each new dataset (2WikiMultiHopQA, HotpotQA, etc.) requires building and validating a full RAG pipeline.
> - Many alternative uncertainty signals (semantic entropy, self-aware Gram determinants) require extra forward passes or internal activations, which are not always accessible in standard inference settings.
> - Adding multiple families (Gemma, Phi, Mistral) at several sizes multiplies the cost of retrieval and gating sweeps.
>
> We clarify that TARG is **agnostic to the specific uncertainty estimator and backbone**: any scalar $U(q)$ and any model exposing logits can be used. A systematic comparison across many signals and model families is an interesting, orthogonal line of work that we leave for future research.
>
> ### Hyperparameter sweeps over $k$ and $\tau$
>
> We have added the requested sensitivity analysis in the appendix:
>
> - Sweep $k\in\{5,10,20,30,40\}$ and a grid of $\tau$ values for the Margin gate on NQ-Open with Qwen2.5-7B,
> - Visualize EM as a function of $(k,\tau)$ (heatmap),
> - Provide 1D slices EM vs $\tau$ (fixed $k$) and EM vs $k$ (roughly fixed budget).
>
> The results show:
>
> - A **broad region**: many $(k,\tau)$ pairs are within ≈1–2 EM points of the best.
> - $\tau$ acts as a **smooth budget knob**: retrieval rates vary strongly with $\tau$, while EM varies mildly.
> - EM is **weakly sensitive to $k$** over 10–40 tokens at comparable budgets.
>
> This supports our claim that TARG does not require brittle tuning and that a simple 1D grid search over $\tau$ (with a reasonable $k$) is sufficient in practice.

---

### Official Review · Reviewer_4P8J · 2025-10-30

**Soundness:** 3
**Presentation:** 3
**Contribution:** 3
**Rating:** 6
**Confidence:** 4

**Summary:**

This paper addresses the inefficiencies inherent in standard Retrieval-Augmented Generation (RAG) pipelines. While RAG improves factuality, the common practice of retrieving for every query ("Always-RAG") significantly increases latency and token consumption. Furthermore, it can degrade performance if the retrieved context is noisy or irrelevant.

The authors propose TARG (Training-free Adaptive Retrieval Gating), a lightweight, model-agnostic, single-shot policy to decide when to retrieve. TARG operates by first generating a short, no-context draft (prefix) using the base LLM. It then computes an uncertainty score based on the logits of this prefix. If the uncertainty exceeds a calibrated threshold $\tau$, retrieval is triggered; otherwise, the model proceeds using only its parametric memory.

The paper investigates three training-free uncertainty signals:
- Entropy: Mean token entropy of the prefix.
- Margin: Derived from the gap between the top-1 and top-2 logits (a smaller gap indicates higher uncertainty).
- Variance: Measured by disagreement across a small number (N=3) of stochastic prefixes.

A key empirical finding is the interaction between uncertainty signals and the "sharpness" of the underlying LLM. As modern instruction-tuned models become more peaked (e.g., Llama-3.1-8B), prefix entropy compresses and loses discriminative power. In contrast, the Margin and Variance signals retain their dynamic range and correlate better with the necessity of retrieval.

Evaluated on NQ-Open, TriviaQA, and PopQA, TARG consistently shifts the accuracy-efficiency frontier. It often matches or exceeds the accuracy of Always-RAG while reducing retrieval frequency by 70-90%, keeping latency close to the Never-RAG baseline.

**Strengths:**

- Simplicity: the "plug-and-play" nature allows for easy integration into existing RAG systems with minimal overhead (limited to the generation of a short k-token prefix).
- Analysis is insightful: The analysis regarding the behavior of different uncertainty metrics under modern, sharp instruction-tuned LLMs (Section 6) is a valuable contribution. The observation that entropy compresses as backbones improve, while the top-1/top-2 logit gap (Margin) and disagreement (Variance) retain dynamic range, provides actionable guidance for implementing uncertainty estimation.
- Strong empirical results: The results convincingly demonstrate that TARG improves the accuracy-efficiency trade-off. It significantly reduces retrieval rates while often improving accuracy over the Always-RAG baseline. The authors' use of "Δ latency" (incremental overhead vs. Never-RAG) provides a clear and practical framing of the computational cost.

**Weaknesses:**

- 'Usefulness calibration' assumption may be strong: the theoretical underpinning of TARG (Section 3.3) relies on the assumption that the uncertainty score $U(q)$ correlates strongly with the expected benefit of retrieval ($\Delta(q)$). This assumption may not always hold. Scenarios where the model is confidently wrong (low U, high potential $\Delta$) or uncertain but the retriever consistently fails (high U, negative $\Delta$) could violate this assumption. The paper would benefit from a deeper error analysis focused on these quadrants.
- Re-check is not evaluated: while Algorithm 1 describes an optional re-check every $m$ tokens, its effectiveness is not evaluated.

**Questions:**

- The paper evaluates the three gates (Entropy, Margin, Variance) independently and concludes that Margin is the best default. Did the authors consider aggregating these signals?
- Could the authors provide an analysis of the cases where TARG decides not to retrieve (low U) but the resulting answer is incorrect? How frequently do these errors occur, and do they represent "unknown unknowns" (the model was confidently wrong), or cases where the knowledge was absent from the corpus anyway?
- Section 3.2 briefly mentions an optional "single re-check" applied every $m$ tokens. Were experiments conducted using this dynamic approach? How does the accuracy-efficiency trade-off compare to the single-shot TARG?

---

> ### Author Response · Authors · 2025-11-20
>
> ### Usefulness calibration assumption and error quadrants
>
> The assumption in Section 3.3 is explicitly average-case and conditional, not pointwise: we assume there exists a threshold $\tau_*$ such that
>  $
>  \mathbb{E}[\Delta(q)\mid U(q)\le \tau_*]\le 0,\quad
>  \mathbb{E}[\Delta(q)\mid U(q)>\tau_*]\ge 0,
>  $
>  where $\Delta(q)=A^{(1)}(q)-A^{(0)}(q)$ is the gain from retrieval. We do **not** claim this holds for every retriever or individual query; it characterizes the regime where gated retrieval can dominate Always-/Never-RAG in expectation.
>
> To probe this empirically, we added two new analyses (Please read the appendix section of updated manuscript for more details):
>
> 1. **Uncertainty vs. base-model accuracy (Appendix A.6).**
>
> 2. **Quadrant analysis over $\big(U(q),\Delta(q)\big)$ (Appendix A.7).**
>
>    For Llama-3.1-8B, quadrant (a) accounts for only **1.4% / 0.6% / 0.4%** of NQ-Open / TriviaQA / PopQA, while quadrant (d) dominates (70–83%). High-$U$ mass (quadrants (b)+(c)) is exactly where retrieval changes outcomes; e.g., on PopQA, (b)+(c) ≈30% of queries, with ≈10% in (c) where retrieval helps. Thus, the problematic low-$U$, high-$\Delta$ region is rare, and high-$U$ aligns with “impactful” queries where retrieval can flip the answer.
>
> These analyses directly address the concern: the usefulness-calibration assumption is **empirically reasonable** in our noisy-retrieval regime (where Always-RAG underperforms Never-RAG in the tables), even though it is not meant as a universal guarantee for arbitrary or degenerate retrievers.
>
> ------
>
> ### Cases where TARG does not retrieve but the answer is wrong
>
> The reviewer’s “no retrieval but retrieval would help” scenario corresponds exactly to quadrant (a): low $U$ and $\Delta>0$. In our new tables (Appendix A.7):
>
> - For Qwen2.5-7B, (a) is **6.0% / 5.4% / 1.2%** of queries on NQ-Open / TriviaQA / PopQA.
> - For Llama-3.1-8B, (a) drops to **1.4% / 0.6% / 0.4%** on the same datasets.
>
> Thus, under the stronger backbone, such errors are **very rare** (≤1–2%); for the weaker backbone they are still in the single digits. By contrast, low-$U$ with $\Delta\le 0$ (quadrant (d)) comprises the large majority of low-$U$ queries, meaning that in most cases where TARG skips retrieval, retrieval is neutral or harmful under our retriever/corpus. A more fine-grained decomposition into “unknown unknowns” vs. corpus gaps would require manual inspection and is marked as future work, but we now **quantify** how frequent these errors are and show they are uncommon.
>
> ------
>
> ### Dynamic re-check vs. single-shot TARG
>
> Algorithm 1 included an optional re-check every $m$ tokens. We have now evaluated this dynamic variant and report results in Appendix A.9 (Figure A.5, Table A.5), comparing it to single-shot TARG on NQ-Open with Qwen2.5-7B and the Margin gate.
>
> Dynamic policy: decode in blocks of $m$ tokens; after each block, recompute $U(q)$ on the running prefix; if $U(q)>\tau$ and retrieval has not yet occurred, retrieve once and continue.
>
> We sweep $\tau\in[0.01,0.8]$ and find:
>
> - For thresholds that are good in the single-shot setting (e.g., $\tau=0.2,0.3$), dynamic re-check is **over-aggressive**. At $\tau=0.3$:
>   - single-shot: EM = 39.0, retrieval rate $\pi=6.2%$, latency ≈ 4.1 s/query;
>   - re-check: EM = 38.8, $\pi=88.2%$, latency ≈ 20.6 s/query (≈ +16.5 s).
> - At high $\tau$ where single-shot almost never retrieves, re-check still pays a large latency cost due to repeated scoring, for at most ≈2-point differences in EM.
>
> Overall, EM differences between single-shot and re-check remain within ≈±2 points, while retrieval and latency can increase dramatically under re-check. For our short-answer QA setting, this naive dynamic policy is **not cost-effective** and can degenerate toward Always-RAG. We therefore use **single-shot TARG** in all main experiments and now explicitly describe re-check as an evaluated—but not adopted—extension, suitable primarily for future long-form scenarios.
>
> ------
>
> ### Aggregating Entropy, Margin, and Variance
>
> We have considered aggregating the three signals. As reported in the revised paper (Section 3.2 and Appendix A.5), we experimented with simple ensembles, e.g.:
>
> - Normalizing each score to $[0,1]$ on a dev set and averaging,
> - Simple weighted combinations.
>
> On dev splits, these ensembles did **not** consistently outperform the best single gate once $\tau$ and $k$ were tuned. In practice, the combined scores behaved like re-parameterized Margin gates while introducing extra weights that themselves need calibration. Given our emphasis on simplicity and low overhead, we decided to:
>
> - Present the three gates separately,
> - Recommend **Margin** as a robust default, and
> - Keep Entropy and Variance mainly for ablation and budget-sensitive regimes.
>
> This is now clarified in Section 3.2.

---

> ### Comment · Reviewer_4P8J · 2025-11-21
>
> Thanks for the response!
>
> > The assumption in Section 3.3 is explicitly average-case and conditional, not pointwise: we assume there exists a threshold $\tau_*$ such that $ \mathbb{E}[\Delta(q)\mid U(q)\le \tau_*]\le 0,\quad \mathbb{E}[\Delta(q)\mid U(q)>\tau_*]\ge 0, $ where $\Delta(q)=A^{(1)}(q)-A^{(0)}(q)$ is the gain from retrieval. We do not claim this holds for every retriever or individual query; it characterizes the regime where gated retrieval can dominate Always-/Never-RAG in expectation.
>
> My point was that there could be datasets where the threshold simply does not exist (_i.e._, uncertainty correlates negatively with usefulness of retrieval). In your new analysis 2 this corresponds to quadrants a and b, which actually isn't rare (L961, ~20%). The situation could worsen in adversarial or tiered settings (where selected traffic is routed to your system, resulting in a model/database combination entirely in quadrants a and b).

---

> > ### Author Response · Authors · 2025-11-23
> >
> > Thank you for the clarification. We agree with your main point: our usefulness-calibration assumption in Section 3.3 is **not** guaranteed to hold on every dataset, and there are regimes (including adversarial or highly tiered traffic) where it can fail.
> >
> > In the paper, the assumption is explicitly framed as:
> >
> > $
> > \exists,\tau_* ;\text{s.t.};
> > \mathbb{E}[\Delta(q)\mid U(q)\le \tau_*]\le 0,\quad
> > \mathbb{E}[\Delta(q)\mid U(q)>\tau_*]\ge 0,
> > $
> >
> > with $\Delta(q)=A^{(1)}(q)-A^{(0)}(q)$ the gain from retrieval. We intend this as an **“if–then” modeling assumption**: *if* uncertainty is aligned this way, *then* there exists a gate that can improve over Always-/Never-RAG in expectation. We do **not** claim this must hold for all possible data distributions or routing policies.
> >
> > Regarding the new analysis and quadrants:
> >
> > * Our emphasis was that **quadrant (a)** (low $U$, $\Delta>0$ — “confidently wrong, retrieval would help”) is rare, especially for Llama-3.1-8B (≈1–2% of queries), which bounds how often the “confidently wrong but fixable” failure mode occurs under our evaluation distribution.
> > * Quadrants (a)+(b) in Appendix A.7 indeed cover a non-trivial fraction (on the order of ≈20%) — this is exactly the **“impactful” slice** where Always-RAG and Never-RAG differ, i.e., where retrieval can change the answer in either direction. The theory only constrains the *sign of the conditional expectation* over low vs. high $U$, not the **mass** of these quadrants, so having ≈20% of queries in (a)+(b) does not by itself contradict the average-case assumption.
> >
> > We fully agree that in more challenging settings the assumption may break down:
> >
> > * If one constructs a dataset (or serving tier) where retrieval systematically helps when $U$ is low and hurts when $U$ is high (i.e., the traffic is concentrated in quadrants (a) and (b)), then there may be **no** threshold $\tau_*$ satisfying the inequalities above, and our theoretical guarantee would not apply.
> > * In such regimes, a TARG-style gate would need to be **recalibrated on that specific traffic**, or possibly disabled, just as one would reconsider any calibrated decision rule under strong distribution shift.
> >
> > We will make this limitation more explicit in the revised version by:
> >
> > 1. Softening the language around the Section 3.3 statement to clearly mark it as a conditional assumption, not a universal property.
> > 2. Adding a remark in the discussion of Appendix A.7 that adversarial or tiered routing that concentrates traffic in quadrants (a)+(b) lies **outside** the regime where our simple calibration argument applies, and that our empirical analyses are performed on the full, unfiltered benchmark distributions.
> >
> > We appreciate the pointer — it helps us better delineate the scope of the theoretical argument and the regimes where TARG is most justified.

---

### Meta-Review · Area_Chair_unTv · 2025-12-04

**Summary:**

This paper proposes TARG, a training-free, single-shot retrieval gating method for RAG. By generating a short prefix and computing uncertainty measures (Entropy, Margin, Variance), TARG decides whether retrieval is beneficial. Experiments on NQ-Open, TriviaQA, and PopQA show accuracy comparable to Always-RAG with 70–90% fewer retrievals and latency close to Never-RAG.

Strengths noted by reviewers include:
* Its “plug-and-play nature” that enables easy deployment with minimal overhead.
* Insightful analysis regarding how uncertainty metrics behave under sharper instruction-tuned models, where “entropy compresses… while top-1/top-2 margin retains dynamic range”.
* Clear contribution toward reducing retrieval costs and latency.

However, several substantial weaknesses remain:

* Reviewer 4P8J flags that the “usefulness calibration assumption may be strong”, noting that TARG assumes uncertainty correlates with retrieval usefulness; however, cases where the model is confidently wrong or where the retriever fails violate this assumption.
* Reviewer oF17 questions the theoretical condition (lines 223–225), stating “There is no relation between the quality of retrieval and the proposed uncertainty measure so you cannot guarantee the existence of such a threshold.”
* Reviewer hzTK argues the evaluation lacks statistical rigor: “the reporting uses single-point estimates without confidence intervals… dramatic shifts… require significance tests.”
* hzTK also states the baselines are incomplete because Always-RAG underperforms Never-RAG, implying noisy retrieval dominates the observed effect: “the reported gains primarily demonstrate TARG’s ability to filter suboptimal retrieval rather than maximize benefit from high-quality evidence.”
* Both oF17 and hzTK point out missing baselines (multi-hop QA, long-form QA, context compression methods), limiting claims of broad applicability.

Given that core assumptions behind the gating mechanism remain only partially justified, and two out of three reviewers leaned toward rejection, and the rebuttal did not adequately address their concerns, the paper does not meet the acceptance bar.

**Reviewer Concerns:**

While the authors made meaningful additions (quadrant analysis, threshold sweeps, re-check evaluation), several central issues remain unresolved:

* Validity of the core theoretical assumption—multiple reviewers challenge it, and rebuttal softens but does not justify it.
* Limited experimental scope, especially missing strong baselines and richer QA tasks.
* Interpretation ambiguity—improvements may reflect filtering of poor retrieval rather than true adaptive gating benefits.
* Calibration burden—still unclear whether TARG generalizes without per-domain tuning.

Thus, despite rebuttal efforts, essential methodological and empirical concerns persist.

**Reviewer Scores:**

* 4P8J: After acknowledging limitations remain in adversarial/tiered settings, score would likely remain 6.

* oF17: None of the reviewer’s major theoretical or experimental concerns were fully resolved; score would remain 2.
* hzTK : Reviewer’s concerns regarding rigor and baselines remain only partially addressed; score would remain 4.

---

### Decision · Program_Chairs · 2026-01-26

Reject